


# The Interactive Stratospheric Aerosol Model Intercomparison Project (ISA-MIP): Motivation and experimental design

Claudia Timmreck[1], Graham W. Mann[2,3], Valentina Aquila[4], Rene Hommel[5,*] Lindsay A. Lee[2], Anja Schmidt[6,7], Christoph Brühl[8] , Simon Carn[9], Mian Chin[10], Sandip S. Dhomse[2], Thomas Diehl[11], Jason M. English[12,13], Michael J. Mills[14], Ryan Neely[2,3], Jianxiong Sheng[15,16], Matthew Toohey[1,17], and Debra Weisenstein[16]

[1]Max-Planck-Institute for Meteorology, Hamburg, Germany
[2]School of Earth and Environment, University of Leeds, Leeds, UK
[3]UK National Centre for Atmospheric Science, University of Leeds, Leeds, UK
[4] American University, Dept. of Environmental Science Washington, DC, USA
[5]Institute of Environmental Physics, University of Bremen, Bremen, Germany
[6] Department of Chemistry, University of Cambridge, Lensfield Road, Cambridge CB2 1EW, UK
[7]Department of Geography, University of Cambridge, Downing Place, Cambridge CB2 3EN, UK
[8] Max-Planck-Institute for Chemistry, Mainz, Germany
[9]Dept Geo Min Eng Sci MTU, Houghton, MI, USA
[10]NASA Goddard Space Flight Center, Greenbelt, MD 20771, USA
[11] Directorate for Sustainable Resources, Joint Research Centre, European Commission, Ispra, Italy
[12] University of Colorado, Cooperative Institute for Research in Environmental Sciences
[13]NOAA Earth Systems Laboratory, Boulder, CO, USA
[14]Atmospheric Chemistry Observations and Modeling, National Center for Atmospheric Research, Boulder, CO, USA
[15] ETHZ, Zürich, Switzerland
[16] John A. Paulson School of Engineering and Applied Sciences, Harvard University, Cambridge, MA, USA
[17]GEOMAR Helmholtz Centre for Ocean Research, Kiel, Germany
[*]now at Hommel & Graf Environmental, Hamburg, Göttingen, Germany

*Correspondence to*: Claudia Timmreck (claudia.timmreck@mpimet.mpg.de)

**Abstract** The Stratospheric Sulfur and its Role in Climate (SSiRC) interactive stratospheric aerosol model intercomparison project (ISA-MIP) explores uncertainties in the processes that connect volcanic emission of sulphur gas species and the radiative forcing associated with the resulting enhancement of the stratospheric aerosol layer. The central aim of ISA-MIP is to constrain and improve interactive stratospheric aerosol models and reduce uncertainties in the stratospheric aerosol forcing by comparing results of standardized model experiments with a range of observations. In this paper we present 4 co-ordinated inter-model experiments designed to investigate key processes which influence the formation and temporal development of stratospheric aerosol in different time periods of the observational record. The "Background" (BG) experiment will focus on microphysics and transport processes under volcanically quiescent conditions, when the stratospheric aerosol is controlled by the transport of aerosols and their precursors from the troposphere to the stratosphere. The "Transient Aerosol Record" (TAR) experiment will explore the role of small- to moderate-magnitude volcanic eruptions, anthropogenic sulphur emissions and transport processes over the period 1998-2012 and their role in the warming hiatus. Two further experiments will investigate the stratospheric sulphate aerosol evolution after major volcanic eruptions. The "Historical Eruptions $SO_2$ Emission Assessment" (HErSEA) experiment will

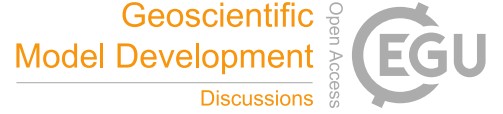



focus on the uncertainty in the initial emission of recent large-magnitude volcanic eruptions, while the
"Pinatubo Emulation in Multiple models" (PoEMS) experiment will provide a comprehensive uncertainty
analysis of the radiative forcing from the 1991 Mt. Pinatubo eruption.

**1 Introduction**

Stratospheric aerosol is an important component of the Earth system, which influences atmospheric radiative
transfer, composition and dynamics, thereby modulating the climate. The effects of stratospheric aerosol on
climate are especially evident when the opacity of the stratospheric aerosol layer is significantly increased after
volcanic eruptions. Through changes in the radiative properties of the stratospheric aerosol layer, volcanic
eruptions are a significant driver of climate variability (e.g. Myhre et al., 2013; Zanchettin et al., 2016). Major
volcanic eruptions inject vast amounts of $SO_2$ into the stratosphere, which is converted into sulphuric acid
aerosol with an e-folding time of about a month. Observations show that the stratospheric aerosol layer remains
enhanced for several years after major eruptions (SPARC, 2006). Such long-lasting volcanic perturbations cool
the Earth's surface by scattering incoming solar radiation and warm the stratosphere by absorption of infrared
solar and long-wave terrestrial radiation which affect the dynamical structure as well as the chemical
composition of the atmosphere (e.g. Robock, 2000; Timmreck, 2012). As the ocean has a much longer memory
than the atmosphere, large volcanic eruptions could have a long lasting impact on the climate system that
extends beyond the duration of the volcanic forcing (e.g., Zanchettin et al., 2012; Swingedouw et al., 2017). The
chemical and radiative effects of the stratospheric aerosol are strongly influenced by its particle size distribution.
Heterogeneous chemical reactions, which most notably lead to substantial ozone depletion (e.g. WMO Ozone
Assessment 2007, chapter 3), take place on the surface of the stratospheric aerosol particles and are dependent
on the aerosol surface area density. Aerosol particle size determines the scattering efficiency of the particles
(e.g. Lacis et al., 1992).and their atmospheric lifetime (e.g., Pinto et al., 1989; Timmreck et al., 2010). Smaller-
magnitude eruptions than 1991 Mt. Pinatubo eruption can also have significant impacts on climate. It is now
established that a series of relatively small magnitude volcanic eruptions caused the increase in stratospheric
aerosol observed between 2000 and 2010 over that period based on ground- and satellite-borne observations
(Vernier et al., 2011b; Neely et al., 2013). Studies have suggested that this increase in stratospheric aerosol
partly counteracted the warming due to increased greenhouse gases over that period (e.g. Solomon et al., 2011;
Ridley et al., 2014; Santer et al., 2015).
Since the 2006 SPARC Assessment of Stratospheric Aerosol Properties Report (SPARC 2006, herein referred as
ASAP2006) the increase in observations of stratospheric aerosol and its precursor gases and in the number of
models which treat stratospheric aerosol interactively, have advanced scientific understanding of the
stratospheric aerosol layer and its effects on the climate (Kremser et al. 2016, herein referred to as KTH2016).
In particular, research findings have given to the community a greater awareness of the role of the tropical
tropopause layer (TTL) as a distinct pathway for transport into the stratosphere, of the interactions between
stratospheric composition and dynamics, and of the importance of moderate-magnitude eruptions in influencing

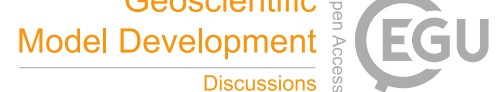



the stratospheric aerosol loading. In addition, over the last decade several new satellite instruments producing
observations relevant to the stratospheric aerosol layer have become operational. For example, we now have a
2002-2012 long record of global altitude-resolved $SO_2$ and OCS measurements provided by the Michelson
Interferometer for Passive Atmospheric Sounding Environmental Satellite (MIPAS Envisat, Höpfner et al.,
2013; 2015; Glatthor et al., 2015). Furthermore aerosol extinction vertical profiles are available from limb-
profiling instruments such as Scanning Imaging Absorption Spectrometer for Atmospheric Chartography
(SCIAMACHY, 2002-2012; Bovensmann et al., 1999; von Savigny et al., 2015), Optical Spectrograph and
InfraRed Imager System (OSIRIS, 2001-present, Bourassa et al., 2007), and Ozone Mapping and Profiler Suite–
Limb Profiler (OMPS-LP, 2011-present, Rault and Loughman, 2013), and from the active sensor lidar
measurements such as Cloud-Aerosol Transport System (CATS, 2015-present, Yorks et al., 2015) and Cloud-
Aerosol Lidar with Orthogonal Polarization (CALIOP, 2006-present, Vernier et al., 2009). Existing
measurements have become more robust, for example by homogenising the observations of aerosol properties
derived from optical particle counter (OPC) and satellite measurements during stratospheric aerosol background
periods (Kovilakam and Deshler, 2015), which previously showed large differences (Thomason et al., 2008).
Other efforts include combining and comparing different satellite data sets (e.g. Rieger et al., 2015). However,
some notable discrepancies still exist between different measurement datasets. For example, Reeves et al. (2008)
showed that aircraft-borne Focused Cavity Aerosol Spectrometer (FCAS) measurements of the particle size
distribution during the late 1990s yield surface area densities a factor 1.5 to 3 higher than that derived from
Stratospheric Aerosol and Gases Experiment (SAGE-II) measurements.
On the modelling side there has been an increasing amount of global three-dimensional stratospheric aerosol
models developed within the last years and used by research teams around the world (KTH2016). The majority
of these global models explicitly simulate aerosol microphysical processes and treat the full life cycle of
stratospheric aerosol, from the initial injection of sulphur containing gases, and their transformation into aerosol
particles, to their final removal from the stratosphere. Several of these models also include the interactive
coupling between aerosol microphysics, atmospheric chemistry, dynamics and radiation.
Given the improvements in observations and modelling of stratospheric aerosol since ASAP2006, we anticipate
further advances in our understanding of stratospheric aerosol by combining the recent observational record
with results from the current community of interactive stratospheric aerosol models. An Interactive
Stratospheric Aerosol Model Intercomparison Project (ISA-MIP) has therefore been developed within the
SSiRC framework. The SPARC activity Stratospheric Sulfur and its Role in Climate (SSiRC) (www.sparc-
ssirc.org) was initiated with the goal of reducing uncertainties in the properties of stratospheric aerosol and
assessing its climate forcing In particular, constraining simulations of historical eruptions with available
observational datasets gives the potential to evaluate and substantially improve the accuracy of the volcanic
forcing datasets used in climate models. This will not only enhance consistency with observed stratospheric
aerosol properties and the underlying microphysical, chemical, and dynamical processes but also improve the
conceptual understanding. The use of such new volcanic forcing datasets has the potential to increase the



reliability of the simulated climate impacts of volcanic eruptions, which have been identified as a major
influence on decadal global mean surface temperature trends in climate models (Marotzke and Forster, 2015).
The first international model inter-comparison of global stratospheric aerosol models was carried out within
ASAP2006 and indicated that model simulations and satellite observations of stratospheric background aerosol
extinction agree reasonably well in the visible wavelengths but not in the infrared. It also highlighted systematic
differences between modelled and retrieved aerosol size, which are not able to detect the Aitken-mode sized
particles (R<50nm) in the lower stratosphere (Thomason et al., 2008, Reeves et al., 2008; Hommel et al. 2011).
While in ASAP2006, only five global two- and three-dimensional stratospheric aerosol models were included in
the analysis, there are nowadays more than 15 global three-dimensional models worldwide available
(KTH2016). No large comprehensive model intercomparison has ever been carried out to identify differences in
stratospheric aerosol properties amongst these new interactive models. The models often show significant
differences in terms of their simulated transport, chemistry, and removal of aerosols with inter-model
differences in stratospheric circulation, radiative-dynamical interactions and exchange with the troposphere
likely to play an important role (e.g. Aquila et al., 2012; Niemeier and Timmreck, 2015). The formulation of
microphysical processes are also important (e.g. English et al. 2013), as are differing assumptions regarding the
sources of stratospheric aerosols and their precursors. A combination of these effects likely explain the large
inter-model differences as seen in Fig. 1 among global stratospheric aerosol models who participated in the
Tambora intercomparison, a precursor to the "consensus volcanic forcings" aspects of the CMIP6 Model
Intercomparison Project on the climatic response to Volcanic forcing (VolMIP, Zanchettin et al., 2016; Marshall
et al., 2017). Even for the relatively recent 1991 Mt. Pinatubo eruption, to reach the best agreement with
observations, interactive stratospheric models have used a wide range of $SO_2$ injections amounts, from as low at
10 Tg of $SO_2$ (Dhomse et al., 2014; Mills et al., 2016) to as high as 20 Tg of $SO_2$ (e.g. Aquila et al., 2012;
English et al., 2013).
Volcanic eruptions are commonly taken as a real-world analogue for hypothesised geoengineering via
stratospheric sulphur solar radiation management (SS-SRM). Indeed many of the assumptions and uncertainties
related to simulated volcanic perturbations to the stratospheric aerosol are also frequently given as caveats
around research findings from modelling studies which seek to quantify the likely effects from SS-SRM (e.g.
National Research Council, 2015 ), the mechanism-steps between sulphur injection and radiative cooling being
common to both aspects (Robock et al., 2013). The analysis of the ISA-MIP experiments we expect to improve
understanding of model sensitivities to key sources of uncertainty, to inform interpretation of coupled climate
model simulations and the next Intergovernmental Panel on Climate Change (IPCC) assessment. It will also
provide a foundation for co-operation to assess the atmospheric and climate changes when the next large-
magnitude eruption takes place.
In this paper, we introduce the new  model intercomparison project ISA-MIP developed within the SSiRC
framework. In section 2 we provide an overview of the current state of stratospheric sulphur aerosol modelling
and its greatest challenges. In section 3 we describe the scopes and protocols of the four model experiments
planned within ISA-MIP. A concluding summary is provided in Section 4.

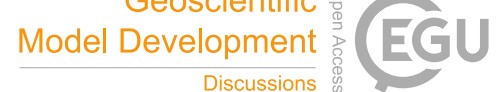



## 2. Modelling stratospheric aerosol; overview and challenges


Before we discuss the current state of stratospheric aerosol modelling and its greatest challenges in detail, we
briefly describe the main features of the stratospheric sulphur cycle. We are aware of the fact that the
stratospheric aerosol layer also contains organics and inclusions of meteoritic dust (Ebert et al., 2016) and, after
volcanic events, also co-exists with volcanic ash (e.g. Pueschel et al., 1994: KTH2016). However, the focus of
the ISA-MIP experiments described here is on comparing to measurements of the overall optical and physical
properties of the stratospheric aerosol layer, which is manly determined by stratospheric aerosol.

### 2.1 The stratospheric aerosol lifecycle


The stratospheric aerosol layer and its temporal and spatial variability are determined by the transport of aerosol
and aerosol precursors in the stratosphere and their modification by chemical and microphysical processes
(Hamill et al., 1997; ASAP2006; KTH2016). Volcanic eruptions can inject sulphur-bearing gases directly into
the stratosphere which significantly enhances the stratospheric aerosol load for years. A number of observations
show that stratospheric aerosol increased over the first decade of the $21^{st}$ century (e.g. Hofmann et al., 2009;
Vernier et al., 2011b; Ridley et al., 2014). Although such increase was attributed to the possible cause of Asian
anthropogenic emission increase (Hofmann et al., 2009), later studies have shown that small-to-moderate
magnitude volcanic eruptions are likely to be the major source of this recent increase (Vernier et al., 2011b;
Neely et al., 2013; Brühl et al., 2015).
A stratospheric source besides major volcanic eruptions is the photochemical oxidation of carbonyl sulphide
(OCS), an insoluble gas mainly inert in the troposphere. Tropospheric aerosols and aerosol precursor also enter
the stratosphere through the tropical tropopause and through convective updrafts in the Asian and North
American Monsoons (Hofmann et al., 2009; Hommel et al., 2011; Vernier et al., 2011a; Bourassa et al., 2012;
Yu et al., 2015). In the stratosphere, new sulphate aerosol particles are formed by binary homogenous nucleation
(Vehkamäki et al., 2002), a process in which sulphuric acid vapour ($H_2SO_4(g)$) and water vapour condense
simultaneously to form a liquid droplet. The condensation of $H_2SO_4(g)$ onto pre-existing aerosol particles and
the coagulation among particles shift the aerosol size distribution to greater radii. This takes place especially
under volcanically perturbed conditions, when the concentrations of aerosol in the stratosphere are higher (e.g.
Deshler et al., 2008).
From the tropics, where most of the tropospheric aerosol enters the stratosphere and the OCS chemistry is most
active, the stratospheric aerosol particles are transported poleward within the large-scale Brewer-Dobson
circulation (BDC) and removed through gravitational sedimentation and cross-tropopause transport in the extra-
tropical regions. Internal variability associated with the quasi-biennial oscillation (QBO) alters the isolation of
the tropical stratosphere and subsequently the extra-tropical transport of the stratospheric aerosol, and modifies
its distribution, particle size, and lifetime (e.g. Trepte and Hitchmann, 1992; Hommel et al., 2015).
In general, under volcanically perturbed conditions with larger amounts of injected $SO_2$, aerosol particles grow
to much larger radii than in volcanic quiescent conditions (e.g. Deshler, 2008). Simulation of extremely large





volcanic sulphur rich eruptions show a shift to particle sizes even larger than observed after the Pinatubo
eruption, and predict a reduced cooling efficiency compared to moderate with moderate sulphur injections (e.g.
Timmreck et al., 2010; English et al., 2013).

### 2.2 Global stratospheric aerosol models, current status and challenges

A comprehensive simulation of the spatio-temporal evolution of the particle size distribution is a continuing
challenge for stratospheric aerosol models. Due to computational constraints, the formation of the stratospheric
aerosol and the temporal evolution of its size distribution are usually parameterized with various degrees of
complexity in global models. The simplest way to simulate the stratospheric aerosol distribution in global
climate models is the mass only (bulk) approach (e.g. Timmreck et al., 1999a; 2003; Aquila et al., 2012), where
only the total sulphate mass is prognostically simulated and chemical and radiative processes are calculated
assuming a fixed typical particle size distribution. More complex methods are size-segregated approaches, such
as the modal approach (e.g. Niemeier et al., 2009; Toohey et al., 2011; Brühl et al., 2012; Dhomse et al., 2014;
Mills et al., 2016), where the aerosol size distribution is simulated using one or more modes, usually of log-
normal shape. The mean radius of each mode of these size distributions varies in time and space. Another
common approach is the sectional method (e.g. English et al., 2011; Hommel et al., 2011; Sheng et al., 2015a;
for ref prior to 2006 see ASAP2006, chapter 5), where the particle size distribution is divided into distinct size
sections. Number and width of the size sections are dependent on the specific model configuration, but are fixed
throughout time and space. Size sections may be defined by an average radius, or by an average mass of
sulphur, and are often spaced geometrically.
The choice of methods has an influence on simulated stratospheric aerosol size distributions and therefore on
radiative and chemical effects. While previous model intercomparison studies in a box model (Kokkola et al.,
2009) or in a two-dimensional framework (Weisenstein et al., 2007) were very useful for the microphysical
schemes, they could not address uncertainties in the spatial transport pattern e.g. transport across the tropopause
and the subtropical transport barrier, or regional/local differences in wet and dry removal. These uncertainties
can only be addressed in a global three-dimensional model framework and with a careful validation with a
variety of observational data.
The June 1991 eruption of Mt. Pinatubo, with the vast net of observations that tracked the evolution of the
volcanic aerosol, provides a unique opportunity to test and validate global stratospheric aerosol models and their
ability to simulate stratospheric transport processes. Previous model studies (e.g. Timmreck et al., 1999b; Aquila
et al., 2012) highlighted the importance of an interactive online treatment of stratospheric aerosol radiative
heating for the simulated transport of the volcanic cloud. A crucial point is the simulation of the tropical
stratospheric aerosol reservoir (i.e., the tropical pipe, Plumb, 1996) and the meridional transport through the
subtropical transport barrier. Some models show a very narrow tropical maximum in comparison to satellite data
(e.g., Dhomse et al. 2014) while others show too fast transport to higher latitudes and fail to reproduce the long
persistence of the tropical aerosol reservoir (e.g. Niemeier et al., 2009; English et al., 2013). Reasons for these
differences need to be understood.

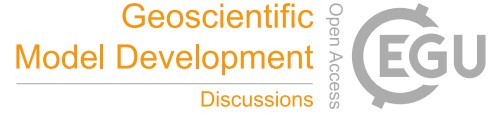

### 3. The ISA-MIP Experiments

Many uncertainties remain in the model representation of stratospheric aerosol. Figure 2 summarizes the main processes that determine the stratospheric sulphate aerosol mass load, size distribution and the associated optical properties. The four experiments in ISA-MIP are designed to address these key processes under a well-defined experiment protocol with prescribed boundary conditions (sea surface temperatures (SSTs), emissions). All simulations will be compared to observations to evaluate model performances and understand model strengths and weaknesses. The experiment "Background" (BG) focuses on microphysics and transport (section 3.1) under volcanically quiescent conditions, when stratospheric aerosol is only modulated by seasonal changes and interannual variability. The experiment "Transient Aerosol Record" (TAR) is addressing the role of time-varying $SO_2$ emission in particular the role of small- to moderate-magnitude volcanic eruptions and transport processes in the upper troposphere – lower stratosphere (UTLS) over the period 1998-2012 (section 3.2). Two further experiments investigate the stratospheric sulphate aerosol size distribution under the influence of large volcanic eruptions. "HErSEA" focuses on the uncertainty in the initial emission characteristics of recent large volcanic eruptions (section 3.3), while "PoEMS" provides an extensive uncertainty analysis of the radiative forcing of the Mt. Pinatubo eruption. In particular the ISA-MIP model experiments aim to address the following questions:

1.  How large is the stratospheric sulphate load under volcanically quiescent conditions, and how sensitive is the simulation of this background aerosol layer to model specific microphysical parameterization and transport? (3.1)

2.  Can we explain the sources and mechanisms behind the observed variability in stratospheric aerosol load since the year 2000? (3.2)

3.  Can stratospheric aerosol observations constrain uncertainties in the initial sulphur injection amount and altitude distribution of the three largest volcanic eruptions of the last 100 years? (3.3)

4.  What is the confidence interval for volcanic forcing of the Pinatubo eruption simulated by interactive stratospheric aerosol models and to which parameter uncertainties are the predictions most sensitive to? (3.4)

Table 1 gives an overview over all ISA-MIP experiments, which are described in detail below. In general each experiment will include several simulations from which only a subset is mandatory (Tier1). The modelling groups are free to choose in which of the experiments they would like to participate, however the BG Tier1 simulation is mandatory for all groups and the entry card for the ISA-MIP intercomparison. All model results will be saved in a consistent format (NETCDF) and made available via http://cera-www.dkrz.de/WDCC/ui, and compared to a set of benchmark observations. More detail technical information about data requests can be found in the supplementary material and on the ISA-MIP webpage: http://www.isamip.eu.

It is mandatory for participating models to run with interactive sulphur chemistry (see review in SPARC ASAP2006) in order to capture the oxidation pathway from precursors to aerosol particles, including aerosol

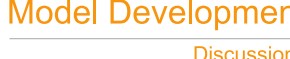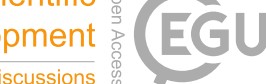

growth due to condensation of $H_2SO_4$. Chemistry Climate Models (CCMs) with full interactive chemistry follow
the Chemistry Climate Initiative (CCMI) hindcast scenario REF-C1 (Eyring et al. 2013,
http://www.met.reading.ac.uk/ccmi/?page_id=11) for the treatment of chemical fields and emissions of
greenhouse gases (GHGs), ozone depleting substances (ODSs), and very short-lived substances (VSLSs). Sea
surface temperatures and sea ice extent are prescribed as monthly climatologies from the MetOffice Hadley
Center Observational Dataset (Rayner et al. 2003). An overview of the boundary conditions is included in the
supplementary material (Table S1). Table S2 reports the inventories to be used for tropospheric emissions of
aerosols and aerosol precursors. Anthropogenic sulphur emissions and biomass burning are taken from the
Monitoring Atmospheric Composition and Climate (MACC)-CITY climatology (Granier et al., 2011). S
emissions from continuously erupting volcanoes are taken into account using Dentener et al. (2006) which is
based on Andres and Kasgnoc (1998). OCS concentrations are fixed at the surface at a value of 510 pptv
(Montzka et al., 2007; ASAP2006). If possible, DMS, dust, and sea salt emissions should be calculated online
depending on the model meteorology. Models considering DMS oxidation should calculate seawater DMS
emissions as a function of wind speed and DMS seawater concentrations. Otherwise, modelling groups should
prescribe for these species their usual emission database for the year 2000. Each group can specify solar forcing
for year 2000 conditions according to their usual dataset.
Modelling groups are encouraged to include a set of passive tracers to diagnose the atmospheric transport
independently from emissions mostly following the CCMI recommendations (Eyring et al., 2013). These tracers
are listed in Table S3 in the supplementary material. Models diagnose aerosol parameters as specified in Tables
S4, S5. Additionally, volume mixing ratios of specified precursors are diagnosed

### 275 3.1 Stratospheric Background Aerosol (BG)

#### 276 3.1.1. Summary of experiment

The overall objective of the BG experiment is to better understand the processes involved in maintaining the
stratospheric background aerosol layer, i.e. stratospheric aerosol not resulting from direct volcanic injections
into the stratosphere. The simulations prescribed for this experiment are time-slice simulations for the year 2000
with prescribed SST including all sources of aerosols and aerosol-precursors except for explosive volcanic
eruptions. The result of BG will be a multi-model climatology of aerosol distribution, composition, and
microphysical properties in absence of volcanic eruptions. By comparing models with different aerosol
microphysics parameterization and simulations of background circulation with a variety of observational data
(Table 2), we aim to assess how these processes impact the simulated aerosol characteristics.

#### 285 3.1.2. Motivation

The total net sulphur mass flux from the troposphere into the stratosphere is estimated to be about 181 Gg S/yr
based on simulations by Sheng et al. (2015a) using the SOCOL-AER model, 1.5 times larger than reported in
ASAP2006 (KTH2016). This estimate, however, could be highly dependent on the specific characteristics of the

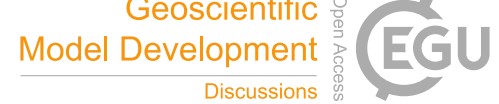



model used, such as strength of convective systems, scavenging efficiency, and occurrence of stratosphere-
troposphere exchange. Therefore, the simulated distribution of stratospheric background aerosol could show,
especially in the lower stratosphere, a very large inter-model variability.
OCS is still considered the largest contributor to the aerosol loadings in the middle stratosphere. Several studies
have shown that the transport to the stratosphere of tropospheric aerosol and aerosol precursors constitutes an
important source of stratospheric aerosol (KTH2016 and references herein) although new in situ measurements
indicate the $SO_2$ flux cross the tropopause is neglible over Mexico and central America (Rollins et al., 2017).
Observations of the Asian Tropopause Aerosol Layer (ATAL, Vernier et al., 2011a) show that, particularly in
the UTLS, aerosol of tropospheric origin can significantly enhance the burden of aerosol in the stratosphere.
This tropospheric aerosol has a more complex composition than traditionally assumed for stratospheric aerosol:
Yu et al. (2015), for instance, showed that carbonaceous aerosol makes up to 50% of the aerosol loadings within
the ATAL. The rate of stratospheric-tropospheric exchange (STE) is influenced by the seasonality of the
circulation and the frequency and strength of convective events in large-scale phenomena such as the Asian and
North American monsoon or in small-scale phenomena such as strong storms. Model simulations by Hommel et
al. (2015) also revealed significant QBO signatures in aerosol mixing ratio and size in the tropical middle
stratosphere (Figure 3). Hence, the model specific implementation of the QBO (nudged or internally generated)
could impact its effects on the stratospheric transport and, subsequently, on the stratospheric aerosol layer.
In this experiment, we aim to assess the inter-model variability of the background stratospheric aerosol layer,
and of the sulphur mass flux from the troposphere to the stratosphere and vice versa. We will exclude changes in
emissions and focus on the dependence of stratospheric aerosol concentrations and properties on stratospheric
transport and stratosphere-troposphere exchange (STE). The goal of the BG experiment aims to understand how
the model-specific transport characteristics (e.g. isolation of the tropical pipe, representation of the QBO and
strength of convective systems) and aerosol parameterizations (e.g. aerosol microphysics and scavenging
efficiency) affect the representation of the background aerosol.

### 313    3.1.3. Experiment setup and specifications

The BG experiment prescribes one mandatory (BG_QBO) and two recommended (BG_NQBO and BG_NAT)
simulations (see Table 3). BG_QBO is a time slice simulation with conditions characteristic of the year 2000[1],
with the goal of understanding sources, sinks, composition, and microphysical characteristics of stratospheric
background aerosol under volcanically quiescent conditions. The time-slice simulation should be at least 20 year
long, after a spin-up period of at least 10 years to equilibrate stratospheric relevant quantities such as OCS
concentrations and age of air. The period seems to be sufficient to study differences in the aerosol properties but
need to extended if dynamical changes e.g. in NH winter variability will be analysed. Modelling groups should
run this simulation with varying QBO, either internally generated or nudged to the 1980-2000 period.

---

[1] To ensure comparability to the AeroCom simulations (http://aerocom.met.no/Welcome.html )





If resources allow, each model should perform the sensitivity experiments BG_NQBO and BG_NAT. The
specifics of these two experiments are the same as for BG_QBO, but BG_NQBO should be performed without
varying QBO[2] and BG_NAT without anthropogenic emissions of aerosol and aerosol precursors, as indicated in
Table S1. The goals of these sensitivity experiments are to understand the effect of the QBO on the background
aerosol characteristics and the contribution of anthropogenic sources to the background aerosol loading in the
stratosphere.

### 3.2 Transient Aerosol Record (TAR)

#### 3.2.1 Summary of experiment

The aim of the TAR (Transient Aerosol Record) experiment is to investigate the relative contributions of
volcanic and anthropogenic sources to the temporal evolution of the stratospheric aerosol layer between 1998
and 2012. Observations show that there is a transient increase in stratospheric aerosol loading, in particular after
the year 2003, with small-to moderate-magnitude volcanic eruptions contributing significantly to this increase
(e.g. Solomon et al., 2011, Vernier et al., 2011b; Neely et al., 2013; Ridley et al. 2014; Santer et al., 2015; Brühl
et al., 2015). TAR model simulations will be performed using specified dynamics, prescribed sea surface
temperature and time-varying $SO_2$ emissions. The simulations are suitable for any general circulation or
chemistry transport models that simulate the stratospheric aerosol interactively and have the capability to nudge
meteorological parameters to reanalysis data. The TAR protocol covers the period from January 1998 to
December 2012, when only volcanic eruptions have affected the upper troposphere and lower stratosphere
(UTLS) aerosol layer with $SO_2$ emissions about an order of magnitude smaller than Pinatubo. Time-varying
surface emission datasets contain anthropogenic and natural sources of sulphur aerosol and their precursor
species. The volcanic $SO_2$ emission inventories contain information of all known eruptions that emitted $SO_2$ into
the UTLS during this period. It comprises the geolocation of each eruption, the amount of $SO_2$ emitted, and the
height of the emissions. $SO_2$ emissions from continuously-degassing volcanoes are also included.

#### 3.2.2 Experiment setup and specifications

Participating models are encouraged to perform up to seven experiments, based on five different volcanic $SO_2$
emission databases (hereafter referred to as VolcDB). Four experiments are mandatory, three other are optional.
The volcanic experiments are compared to a reference simulation (noVolc) that does not use any of the volcanic
emission databases, but emissions from continuously-degassing volcanoes. The aim of the reference simulation
is to simulate the non-volcanically perturbed state of the stratospheric aerosol layer. In contrast to the
experiment protocol BG (Section 3.1), here time-varying surface boundary conditions (SST/SIC) are applied,
whereas BG intercompares model simulations under climatological mean conditions and uses constant 2000
conditions.

---

[2] Models with an internal generated QBO might nudge the tropical stratospheric winds.



An overview of the volcanic emission inventories is given in Table 4 and in Figure 4 VolcDB1/2/3 are new
compilations (Bingen et al., 2017; Neely and Schmidt, 2016; Carn et al., 2016), whereas a fourth inventory
(VolcDB4; Diehl et al., 2012), provided earlier, for the AeroCom community modelling initiative, is optional.
The databases use $SO_2$ observations from different sources and apply different techniques for the estimation of
injection heights and the amount of emitted $SO_2$. The 4 inventories are provided in the form of tabulated point
sources, with each modelling group t
o translate emitted $SO_2$ mass for each eruption into model levels spanning the upper and lower emission
altitudes. If modelling groups prefer not to use point sources, we additionally offer VolcDB1_3D which
provides a series of discrete 3D gridded $SO_2$ injections at specified times. In both versions of VolcDB1, the
integral $SO_2$ mass of each injection is consistent.
We recommend performing one additional non-mandatory experiment in order to quantify and isolate the effects
of 8 volcanic eruptions that either had a statistically significant effect on, for instance, tropospheric temperatures
(Santer et al., 2014, 2015) or emitted significant amounts of $SO_2$ over the 1998 to 2012 time period. This
experiment uses a subset of volcanic emissions (VolcDBSUB), that were derived based on the average mass of
$SO_2$ emitted using VolcDB1, VolcDB2, and VolcDB3 for the following eruptions: 28 January 2005 Manam
(4.0S, Papua New Guinea), 7 October 2006 Tavurvur (4.1 S, Papua New Guinea), 21 June 2009 Sarychev,
(48.5° N, Kyrill, UDSSR) 8 November 2010 Merapi (7.3° S, Java, Indonesia), and 21 June 2011 Nabro (13.2°
N, Eritrea). In addition the eruptions of Soufriere Hills (16.4° N, Monserrat) on 20 May 2006, Okmok (53.3° N,
Alaska) on 12 July 2008 and Kasatochi (52.1° N, Alaska) on 7 August 2008 are considered although these are
not discernible in climate proxy (Kravitz and Robock, 2010; Santer et al., 2014; 2015).
Summarising the number of experiments to be conducted within TAR: four are mandatory (noVolc,
VolcDB1/2/3), one additional is recommended (VolcDBSUB) and two others are optional (VolcDB4 and
VolcDB1_3D; see Table 5 for an overview).
***Volcanic $SO_2$ Emission Databases***
VolcDB1 (Bingen et al., 2017 and Table S6) are updates from Brühl et al. (2015) using satellite data of MIPAS
and OMI. For TAR, VolcDB1 has been extended based on data from Global Ozone Monitoring by Occultation
of Stars (GOMOS), SAGE II, Total Ozone Mapping Spectrometer (TOMS), and the Smithsonian database. The
optionally provided VolcDB1_3D data set, contains volume mixing ratio distributions of the injected $SO_2$ on a
T42 Gaussian grid with 90 levels. VolcDB2 (Mills et al., 2016; Neely and Schmidt, 2016) contains volcanic $SO_2$
emissions and plume altitudes for eruptions between that have been detected by satellite instruments including
TOMS, OMI, OMPS, Infrared Atmospheric Sounding Interferometer (IASI), Global Ozone Monitoring
Experiment (GOME/2), Atmospheric Infrared Sounder (AIRS), Microwave Limb Sounder (MLS) and the
MIPAS instrument. The database is compiled based on published estimates of the eruption source parameters
and reports from the Smithsonian Global Volcanism Program (http://volcano.si.edu/), NASA's Global Sulfur
Dioxide Monitoring website (http://so2.gsfc.nasa.gov/) as well as the Support to Aviation Control Service
(http://sacs.aeronomie.be/). The tabulated point source database also includes volcanic eruptions that emitted



$SO_2$ into the troposphere only, as well as direct stratospheric emissions and has been used and compared to
observations in Mills et al. (2016) and Solomon et al. (2016).
VolcDB3 uses the most recent compilation of the volcanic degassing data base of Carn et al. (2016).
Observations from the satellite instruments TOMS, the High-resolution Infrared Sounder (HIRS/2), AIRS, OMI,
MLS, IASI and OMPS are considered, measuring in the UV, IR and microwave spectral bands. Similar to
VolcDB1/2, VolcDB3 also includes tropospheric eruptions.
Historically VolcDB4 is an older dataset, which relies on information from OMI, the Global Volcanism
Program (GVP), and other observations from literature, covering time period from 1979 to 2010. In contrast to
the other inventories, VolcDB4 has previously been applied by a range of models within the AeroCom,
community (http://aerocom.met.no/emissions.html; Diehl et al., 2012, Dentener et al., 2006). Hence, it adds
valuable information to the TAR experiments because it allows estimating how the advances in observational
methods impact modelling results. It should be noted that VolcDB4 already contains the inventory of Andres
and Kasgnoc (1998) for S emissions from continuously erupting volcanoes and should not be allocated twice
when running this experiment.
*Boundary Conditions, Chemistry and Forcings*
To reduce uncertainties associated with model differences in the reproduction of synoptic and large-scale
transport processes, models are strongly encouraged to perform TAR experiments with specified dynamics,
where meteorological parameters are nudged to a reanalysis such as the ECMWF ERA-Interim (Dee et al.,
2011). This allows models to reasonably reproduce the QBO and planetary wave structure in the stratosphere
and to replicate as closely as possible the state of the BDC in the simulation period. Nudging also allows
comparing directly to available observations of stratospheric aerosol properties (Table 2), such as the extinction
profiles and AOD, and should enable the models to simulate the Asian tropopause layer (ATAL; Vernier et al.,
2011a; Thomason and Vernier, 2013), which, so far, has been studied only by very few global models in great
detail (e.g. Neely et al., 2014; Yu et al., 2015).

**3.3. Historical Eruption SO2 Emission Assessment" (HErSEA)**
**3.3.1 Summary of experiment**
This Historical Eruption $SO_2$ Emission Assessment (HErSEA) experiment will involve each participating model
running a limited ensemble of simulations for each of the three largest volcanic perturbations to the stratosphere
in the last 100 years: 1963 Mt. Agung, 1982 El Chichón and 1991 Mt. Pinatubo.
The main aim is to use a wide range of stratospheric aerosol observations to constrain uncertainties in the $SO_2$
emitted for each eruption (amount, injection height). Several different aerosol metrics will be intercompared to
assess how effectively the emitted $SO_2$ translates into perturbations to stratospheric aerosol properties and
simulated radiative forcings across interactive stratospheric aerosol CCMs with a range of different
complexities. Whereas the TAR simulations (see section 3.2) use specified dynamics, and are suitable for

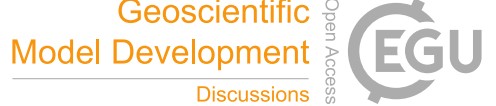



425 chemistry transport models, for this experiment, simulations must be free-running with radiative coupling to the

426 volcanically-enhanced stratospheric aerosol, thereby ensuring the composition-radiation-dynamics interactions

427 associated with the injection are resolved. We are aware that this specification inherently excludes chemistry

428 transport models, which must impose atmospheric dynamics. However, since the aim is to apply stratospheric

429 aerosol observations in concert with the models to re-evaluate current best-estimates of the $SO_2$ input, and in

430 light of the first order impact the stratospheric heating has on hemispheric dispersion from these major eruptions

431 (e.g. Young, R. E. et al., 1994), we assert that this apparent exclusivity is entirely justified in this case.

432 As well as analysing and evaluating the individual model skill and identifying model consensus and

433 disagreement for these three specific eruptions, we also seek to learn more about major eruptions which

434 occurred before the era of satellite and in-situ stratospheric measurements. Our understanding of the effects

435 from these earlier eruptions relies on deriving volcanic forcings from proxies such as sulphate deposition to ice

436 sheets (Gao et al., 2007; Sigl et al., 2015; Toohey et al., 2013), from photometric measurements from

437 astronomical observatories (Stothers, 1996, 2001) or from documentary evidence (Stothers, 2002; Stothers and

438 Rampino, 1983; Toohey et al., 2016a).

439 **3.3.2 Motivation**

440 In the days following the June 1991 Pinatubo eruption, satellite $SO_2$ measurements show (e.g. Guo et al.,

441 2004a) that the peak gas phase sulphur loading was 7 to 11.5 Tg [S] (or 14 -23 Tg $SO_2$). The chemical

442 conversion to sulphuric aerosol that occurred in the tropical reservoir over the following weeks, and the

443 subsequent transport to mid- and high-latitudes, caused a major enhancement to the stratospheric aerosol layer.

444 The peak particle sulphur loading, through this global dispersion phase, reached only around half that in the

445 initial $SO_2$ emission , the maximum particle sulphur loading measured as 3.7 to 6.7 Tg [S] (Lambert et al., 1993;

446 Baran and Foot, 1994), based on an aqueous sulphuric acid composition range of 59 to 77% by weight (Grainger

447 et al., 1993).

448 Whereas some model studies with aerosol microphysical processes find consistency with observations for $SO_2$

449 injection values of 8.5 Tg S (e.g., Niemeier et al., 2009; Toohey et al., 2011; Brühl et al., 2015), several recent

450 microphysical model studies (Dhomse et al., 2014; Sheng et al. 2015a; Mills et al., 2016) find best agreement

451 for an injected sulphur amount at, or even below, the lower end of the range from the satellite $SO_2$

452 measurements. Model predictions are known to be sensitive to differences in assumed injection height (e.g.

453 Sheng et al., 2015b, Jones et al., 2016) and whether models resolve radiative heating and "self-lofting" effects

454 also affects subsequent transport pathways (e.g. Young, R. E. et al., 1994; Timmreck et al. 1999b; Aquila et al.,

455 2012). Another potential mechanism that could explain part of the apparent model-observation discrepancy is

456 that a substantial proportion of the sulphur may have been removed from the plume in the first months after the

457 eruption due to accommodation onto co-emitted ash/ice (Guo et al., 2004b) and subsequent sedimentation.

458 This ISA-MIP experiment will explore these issues further, with the participating models carrying out co-

459 ordinated experiments of the three most recent major eruptions, with specified common $SO_2$ amounts and

460 injection heights (Table 6). This design ensures the analysis can focus on key inter-model differences such as





stratospheric circulation/dynamics, the impacts from radiative-dynamical interactions and the effects of aerosol
microphysical schemes.  Analysing how the vertical profile of the enhanced stratospheric aerosol layer evolves
during global dispersion and decay, will provide a key indicator for why the models differ, and what are the key
driving mechanisms. For all three major eruptions, we have identified key observational datasets (Table 7) that
will provide benchmark tests to evaluate the vertical profile, covering a range of different aerosol metrics.
**3.3.3 Experiment setup and specifications**
Each modelling group will run a mini-ensemble of transient AMIP-type runs for the 3 eruptions with upper and
lower bound $SO_2$ emissions and 3 different injection height settings: two shallow (e.g. 19-21 km and 23-25 km)
and one deep (e.g. 19-25 km) (see Table 7). The seasonal cycle of the Brewer Dobson circulation affects the
hemispheric dispersion of the aerosol plume (e.g. Toohey et al., 2011) and the phase of the QBO is also known
to be key control for tropical eruptions (e.g. Trepte and Hitchman, 1992). To quantify the contribution of the
tracer transport, a passive tracer Volc (Table S3) will be additionally initialized. Note since the AMIP-type
simulations will be transient, prescribing time-varying sea-surface temperatures, the models will automatically
match the surface climate state (ENSO, NAO) through each post-eruption period. Where possible, models
should re-initialise (if they have internally generated QBO) or use specified dynamics approaches (e.g. Telford
et al., 2008) to ensure the model dynamics is consistent with the QBO evolution through the post-eruption
period. General circulation models should use GHG concentrations appropriate for the period and models with
interactive stratospheric chemistry should ensure the loading of Ozone Depleting Substances (ODSs) matches
that for the time period.
Table 8 shows the settings for the $SO_2$ injection for each eruption. Note that experience of running interactive
stratospheric aerosol simulations shows that the vertical extent of the enhanced stratospheric aerosol will be
different from the altitude range in which the $SO_2$ is injected. So, these sensitivity simulations will allow to
assess the behaviour of the individual models with identical settings for the $SO_2$ injection.
For these major eruptions, where the perturbation is much larger than in TAR, model diagnostics include AOD
and extinction at multiple wavelengths and heating rates (K/day) in the lower stratosphere to identify the
stratospheric warming induced by simulated volcanic enhancement, including exploring compensating effects
from other constituents (e.g. Kinne et al., 1992). To allow the global variation in size distribution to be
intercompared, models will also provide 3D-monthly effective radius, with also cumulative number
concentration at several size-cuts for direct comparison to balloon measurements. Examining the co-variation of
the particle size distribution with variations in extinction at different wavelengths will be of particular interest in
relation to approaches used to interpret astronomical measurements of eruptions in the pre-in-situ era (Stothers,
1996, 2001). A 3-member ensemble will be submitted for each different injection setting.

**3.4. Pinatubo Emulation in Multiple models" (PoEMs)**





### 3.4.1 Summary of experiment


The PoEMS experiment will involve each interactive stratospheric aerosol model running a perturbed parameter
ensemble (PPE) of simulations through the 1991-1995 Pinatubo-perturbed period. Variation-based sensitivity
analysis will derive a probability distribution function (PDF) for each model's predicted Pinatubo forcing,
following techniques applied successfully to quantify and attribute sources of uncertainty in tropospheric aerosol
forcings (e.g. Carslaw et al., 2013). The approach will teach us which aspects of the radiative forcing from
major eruptions is most uncertain, and will enable us to identify how sensitive model predictions of key features
(e.g. timing and value of peak forcing and decay timescales) are to uncertainties in several model parameters.
By comparing the time-signatures of different underlying aerosol metrics (mid-visible AOD, effective radius,
particle number) between models, and crucially also against observations, may also help to reduce the natural
forcing uncertainty, potentially thereby making the next generation of climate models more robust.

### 3.4.2 Motivation


The sudden global cooling from major eruptions is a key signature in the historical climate record and a natural
global warming signature occurs after peak cooling as volcanic aerosol is slowly removed from the stratosphere.
Quantitative information on the uncertainty range of volcanic forcings is therefore urgently needed. The amount
of data collected by satellite-, ground-, and air-borne instruments in the period following the 1991 eruption of
Mount Pinatubo (see e.g. section 3.3.2, Table 7) provides an opportunity to test model capabilities in simulating
large perturbations of stratospheric aerosol and their effect on the climate. Recent advances in quantify
uncertainty in climate models (e.g. Rougier et al., 2009;Lee at al. 2011) involve running ensembles of
simulations to systematically explore combinations of different external forcings to scope the range of possible
realisations. There are now a large number of general circulation models (GCMs) with prognostic aerosol
modules, which tend to assess the stratospheric aerosol perturbation through the Pinatubo-perturbed period (see
Table 9). Although these different models achieve reasonable agreement with the observations, this consistency
of skill is achieved with considerable diversity in the values assumed for the initial magnitude and distribution
of the $SO_2$ injection. The $SO_2$ injections prescribed by different models range from 5Tg-S to 10 Tg-S, and the
upper edge of the injection altitude varies among models from as low as 18km to as high as 29km, as shown in
Table 9. Such simulations also differ in the choice of the vertical distribution of $SO_2$ injection (e.g. uniform,
Gaussian, or triangular distributions) and the horizontal injection area (one to several grid boxes). The fact that
different choices of injection parameters lead to similar results in different models points to differences in the
models' internal treatment of aerosol evolution. Accurately capturing microphysical processes such as
coagulational, growth and subsequent rates of sedimentation has been shown to be important for volcanic
forcings (English et al., 2013), but some studies (e.g. Mann et al., 2015) identify that these processes interplay
also with aerosol-radiation interactions, the associated dynamical effects changing the fate of the volcanic
sulphur and its removal into the troposphere. The PoEMS experiment will specifically assess this issue by
adjusting the rate of specific microphysical processes in each model simultaneously with perturbations to SO2

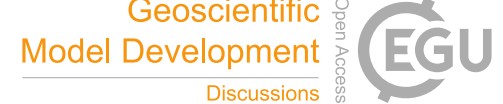



emission and injection height, thereby assessing the footprint of their influence on subsequent volcanic forcing
in different complexity aerosol schemes and the relative contribution to uncertainty from emissions and
microphysics.

### 3.4.3 Experiment setup and specifications

For each model, an ensemble of simulations will be performed varying $SO_2$ injection parameters and a selection
of internal model parameters within a realistic uncertainty distribution. A maximin Latin hypercube sampling
strategy will be used to define parameter values to be set in each PPE member in order to obtain good coverage
of the parameter space. The maximin Latin hypercube is designed such that the range of every single parameter
is well sampled and the sampling points are well spread through the multi-dimensional uncertainty space – this
is achieved by splitting the range of every parameter into N intervals and ensuring that precisely one point is in
each interval in all dimensions, where N is the total number of model simulations, and the minimum distance
between any pair of points in all dimensions is maximised. Fig. 6 shows the projection onto two dimensions of a
Latin hypercube built in 8 dimensions with 50 model simulations. The size of the Latin hypercube needed will
depend on the number of model parameters to be perturbed; the number of simulations to be performed will be
equal to seven times the number of parameters. All parameters are perturbed simultaneously in the Latin
hypercube.
In order to be inclusive of modelling groups with less computing time available, and different types of aerosol
schemes, we define 3 options of experimental design with different numbers of perturbed parameters and thus
simulation ensemble members. The 3 options involve varying all 8 (standard set), 5 (reduced set), or 3
(minimum set) of the list of uncertain parameters, resulting in ensembles of 64 (standard), 40 (reduced) or 24
(minimum) PPE members. The parameters to be varied are shown in Table 10, and include variables related to
the volcanic injection, such as its magnitude, height, latitudinal extent, and composition, and to the life cycle of
the volcanic sulphate, such as the sedimentation rate, its microphysical evolution, and the $SO_2$ to $SO_4^{2-}$
conversion rate.
Prior to performing the full PPE, modelling groups are encouraged to run "One-At-a-Time" (OAT) test runs
with each of the process parameters increased/decreased to its maximum/minimum value. Submission of these
OAT test runs is encouraged (following the naming convention in Table 11) because as well as being an
important check that the model parameter-scaling is being implemented as intended, the results will also enable
intercomparison of single-parameter effects between participating models ahead of the full ensemble. That this
restriction to the parameter-scalings is operational is an important preparatory exercise and will need to have
been verified when running the OAT test runs.
Once a modelling group has performed the PPE of simulations as defined by the Latin hypercube a statistical
analysis will be performed. Emulators for each of a selection of key metrics will be built, following the
approach described by Lee et al. (2011), to examine how the parameters lead to uncertainty in key features of
the Pinatubo-perturbed stratospheric aerosol. The emulator builds a statistical model between the ensemble



design and the key model output and once validated allows sampling of the whole parameter space to derive a
PDF of each key model output.
Variance-based sensitivity analysis will then be used to decompose the resulting probability distribution into its
sources providing information on the key sources of uncertainty in any model output. The two sensitivity indices
of interest are called the main effect and the total effect. The main effect measures the percentage of uncertainty
in the simulated metric due to each parameter-variation individually. The total effect measures the percentage of
uncertainty in the key model output due to each parameter, including the additional contribution from its
interaction with other uncertain parameters. The sources of model parametric uncertainty (i.e. the sensitivity
indices) will be identified for each model with discussion with each group to check the results. By then
comparing the sensitivity to the uncertain parameters across the range of participating models, we will learn
about how the model's differing treatment of aerosol processes, and the inherent dynamical and chemical
processes resolved in the host model, together determine the uncertainty in its predicted Pinatubo radiative
forcings.
The probability distribution of observable key model outputs will also be compared to observations, in order to
constrain the key sources of uncertainty and thereby reduce the parametric uncertainty in individual models. The
resulting model constraints will be compared between models providing quantification of both parametric
uncertainty and structural uncertainty for key variables such as AOD, effective radius and radiative flux
anomalies. This sensitivity analysis will also identify the variables for which better observational constraints
would yield the greatest reduction in model uncertainties.

**4. Conclusions**
The ISA-MIP experiments will improve understanding of stratospheric aerosol processes, chemistry, and
dynamics, and constrain climate impacts of background aerosol "variability", small volcanic eruptions, and
large volcanic eruptions. The experiments will also help to resolve some disagreements amongst global aerosol
models, for instance the difference in volcanic $SO_2$ forcing efficacy for Pinatubo (see section 3.3.2). The results
of this work will help constrain the contribution of stratospheric aerosols to the early 21st century global
warming hiatus period, the effects from hypothetical geoengineering schemes, and other climate processes that
are influenced by the stratosphere. Overall they provide an excellent opportunity to answer some of these
questions as part of the greater WCRP SPARC and CMIP6 efforts.
As well as identifying areas of agreement and disagreement among the different complexities of models in top-
level comparisons focussing on fields such as zonal-mean mid-visible AOD and extinction profiles in different
latitudes, we also intend to explore relationships between key parameters. For example, how does sulphate
deposition to the polar ice sheets relate to volcanic forcing in the different interactive stratospheric aerosol
models that predict the transport and sedimentation of the particles? Or how do model "spectral extinction
curves" evolve through the different volcanically-perturbed periods and how do they relate to simulated
effective radius compared to the theoretical approach to derive effective radius from Stothers (1997; 2001).
There is considerable potential to apply the model uncertainty analysis to make new statements to inform our

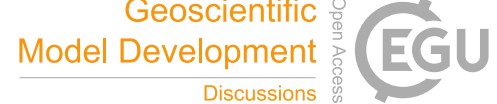



confidence of volcanic forcings derived from ice core and astronomical measurements for eruptions before the
in-situ measurement era.

**Code and data availability**
The model output from the all simulations described in this paper will be distributed through the World Data
climate Center https://www.dkrz.de/up/systems/wdcch with digital object identifiers (DOIs) as-signed. The
model output will be freely accessible through this data portal after registration.

**Authorcontributions.**
CT, GWM VA, RH, LAL, AS, CB, SC MC, SSD, TD, JME, MJM, RN, JXS, MT and D.W designed the
experiments. CT and GWM coordinated the writing, and drafted the manuscript. All authors have contributed to
the writing and have approved of the final version of the manuscript.

**Competing interests.**
The authors declare that they have no conflict of interest.
**Acknowledgements**
The authors thank their SSiRC colleagues for continuing support and discussion. We acknowledge the scientific
guidance (and sponsorship) of the World Climate Research Programme to motivate this work, to be coordinated
in the framework of SPARC. C. Timmreck, M. Toohey and R. Hommel acknowledge support from the German
federal Ministry of Education (BMBF), research programmes "MiKlip"
(FKZ:01LP130A(CT):/01LP1130B(MT)), and ROMIC-ROSA (FKZ: 01LG1212A (RH)). C. Timmreck is also
supported by the European Union project StratoClim (FP7-ENV.2013.6.1-2). C. Brühl's PhD student S.
Schallock, who contributed to the compilation of the volcano inventory, is also supported by StratoClim. A.
Schmidt was funded by an Academic Research Fellowship from the School of Earth and Environment,
University of Leeds and NERC grant NE/N006038/1. M. Toohey acknowledges support by the Deutsche
Forschungsgemeinschaft (DFG) in the framework of the priority programme "Antarctic Research with
comparative investigations in Arctic ice areas" through grant TO 967/1-1. The National Center for Atmospheric
Research is funded by the National Science Foundation. Lindsay Lee is a Leverhulme Early Career Fellow
funded under the Leverhulme Trust grant ECF-2014-524.

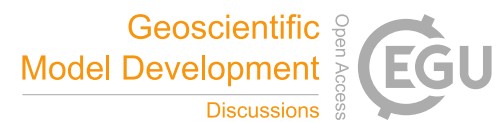

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





**Tables**

| Experiment | Focus | Number of specific experiments | Years per experiment | Total years [A] | Knowledge-gap to be addressed |
|---|---|---|---|---|---|
| **Background Stratospheric Aerosol [BG]** | Stratospheric sulphur budget in volcanically quiescent conditions | 1 mandatory + 2 recommended | 20 | 20(60) | **20 year climatology** to understand **sources and sinks** of stratospheric background aerosol, assessment of sulfate aerosol load under **volcanically quiescent conditions** |
| **Transient Aerosol Record [TAR]** | Transient stratospheric aerosol properties over the period 1998 to 2012 using different volcanic emission datasets | 4 mandatory +3 optional experiments recommended are 5 (see also Table 4 ) | 15 | 60 (75,105) | Evaluate models over the period 1998-2012 with different volcanic emission data sets **Understand drivers and mechanisms for observed stratospheric aerosol changes since 1998** |
| **Historic Eruption SO₂ Emission Assessment [HErSEA]** | Perturbation to stratospheric aerosol from SO₂ emission appropriate for 1991 Pinatubo, 1982 El Chichón,1963, Agung | for each (x3) eruption (Control, median and 4 (2x2) of hi/lo deep/shallow (see also Table 6) | 4 recom. 6 | 180 (270) | Assess how injected SO₂ propagates through to radiative effects for different historical major tropical eruptions in the different interactive stratospheric aerosol models  Use stratospheric aerosol measurements to constrain uncertainties in emissions and gain new observationally-constrained volcanic forcing and surface area density datasets  **Explore the relationship between volcanic emission uncertainties and volcanic forcing uncertainties** |
| **Pinatubo Emulation in Multiple Models [PoEMS][B]** | Perturbed parameter ensemble of runs to quantify uncertainty in each model's predictions | Each model to vary , 5 or 3 of 8 parameters (7 per parameter = 56 35 or 21) | 5 per parameter | 280, 175 or 105 (8, 5 or 3) | Intercompare Pinatubo perturbation to strat- aerosol properties with full uncertainty analysis over PPE run by each model.  Quantify sensitivity of predicted Pinatubo perturbation stratospheric aerosol properties and radiative effects to uncertainties in injection settings and model processes  Quantify and intercompare sources of uncertainty in simulated Pinatubo radiative forcing for the different complexity models. |

[A] Each model will need to include an appropriate initialization and spin-up time for each ensemble member (~3-6 years depending on model
configuration).
[B] Note, that we are aware that some of the structural parameter variations in PoEMS will introduce some inherent drift in stratospheric
aerosol properties for the background control run. However, initial test runs suggest the effect will be much larger for the volcanic
perturbation. We therefore expect the effect of the control-drift on derived radiative forcings to be small. Models running tropospheric and
stratospheric aerosol interactively will need to restrict the parameter scaling to the stratosphere.
**Table 1 General overview of the SSIRC ISA-MIP experiments.**



| Measurement/Platform | Time period 1998-2014 | Reference |
|---|---|---|
| SO$_2$ profile/MLS | 2004-2011 | Pumphrey et al., 2015 |
| SO$_2$ profile/MIPAS | 2002-2012 | Höpfner et al., 2013; 2015 |
| Aerosol extinction profile, size/SAGE II | 1998-2005 | Russell and McCormick, 1989 |
| Aerosol extinction profile, size/OSIRIS | 2001-2011 | McLinden et al., 2012; Rieger et al., 2015 |
| Aerosol extinction profile/GOMOS | 2002-2021 | Vanhellemont et al., 2010 |
| Aerosol extinction profile/SCIAMACHY | 2002-2012 | Taha et al., 2011; von Savigny et al. 2015 |
| Aerosol extinction profile/CALIOP | 2006-2011 | Vernier et al., 2009, 2011a,b |
| Aerosol extinction or AOD merged products | 1998-2011 | Rieger et al., 2015 |
| AOD from AERONET and lidars | | Ridley et al., 2014 |
| Surface area density | | Kovilakam and Deshler, 2015 Eyring et al. (2013) |


**Table 2: List of stratospheric aerosol and SO$_2$ observations available for the BG and TAR time period.**

| Exp- Name | Specific description / Volcanic emission | Period | Ensemble Size | Years per member | Tier |
|---|---|---|---|---|---|
| BG_QBO | Background simulation | Time slice year-2000 monthly-varying with internal or nudged QBO | 1 | 20 | 1 |
| BG_NQBO | Perpetual easterly phase of the QBO for the whole simulation | Time slice year-2000 monthly varying without QBO | 1 | 20 | 2 |
| BG_NAT | Only natural sources of aerosol (including biomass burning) | Time slice year-2000 monthly varying with internal of nudged QBO (when possible) | 1 | 20 | 2 |

**Table 3: Overview of BG experiments.**

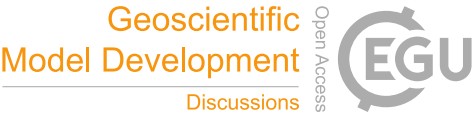



| Volcanic Database | VolcDB1 | VolcDB2 | VolcDB3 | VolcDB4 | VolcDBSUB | VolcDB1_3D |
|---|---|---|---|---|---|---|
| **Covering period** | Dec/1997 - Apr/2012 | Jan/1990 - Dec/2014 | 1978-2014 | 1979-2010 | | Dec/1997- Apr/2012 |
| **Observational data sets** | MIPAS,.GOMOS, SAGEII, TOMS, OMI | OMI, OMPS, IASI, TOMS, GOME/2, , AIRS, MLS, MIPAS | TOMS, HIRS/2, AIRS, OMI, MLS, IASI and OMPS | TOMS, OMI | | MIPAS,.GOMOS, SAGEII, TOMS, OMI |
| **Reference** | Brühl et al. (2015), Bingen et al. (2017), Table S6 | Mills et al. (2016, Neely and Schmidt (2016)) | Carn et al. (2016) | Diehl et al.,(2012), AeroCom-II HCA0 v1/v2, http://aerocom.met.no/emissions.html | Subset of 8 volcanoes Contains SO$_2$ emissions and plume altitudes averaged over the 3 mandatory databases, **details are given in the appendix.** | 3D netCDF Brühl et al. (2015), Bingen et al. (2017), Table S.6 |


**Table 4: Overview of volcanic emission data sets for the different TAR experiments. Sensor acronyms: (MIPAS:**
**Michelson Interferometer for Passive Atmospheric Sounding; GOMOS: Global Ozone Monitoring by Occultation of**
**Stars TOMS: Total Ozone Mapping Spectrometer; OMI: Ozone Monitoring Instrument; OMPS: Ozone Mapping**
**and Profiler Suite; IASI: Infrared Atmospheric Sounding Interferometer; GOME: Global Ozone Monitoring**
**Experiment; AIRS: Atmospheric Infrared Sounder; MLS: Microwave Limb Sounder; HIRS: High-resolution**
**Infrared Radiation Sounder; (References to the observational data and emission sources included are given in the**
**reference paper and for VolcDB1(_3D) also in Table S2.1. VolcDB1_3D is a three-dimensional database, containing**
**the spatial distributions of the injected SO$_2$ as initially observed by the satellite instruments. In both versions of**
**VolcDB1, the integral SO$_2$ mass of each injection is consistent.**







| Exp- Name | Volcanic Database Name | Specific description | Period | Years per member | TiER |
|---|---|---|---|---|---|
| TAR_base | -- | No sporadically erupting volcanic emission | Transient 1998-2012 monthly-varying | 15 | 1 |
| TAR_db1 | **VolcDB1** | Volcanic emission data set (Bruehl et al., 2015 and updates) | Transient 1998-2012 monthly-varying | 15 | 1 |
| TAR_db2 | **VolcDB2** | Volcanic emission data set (Mills et al. 2016) | Transient 1998-2012 monthly-varying | 15 | 1 |
| TAR_db3 | **VolcDB3** | Volcanic emission data set (Carn et al. 2016) | Transient 1998-2012 time-varying | 15 | 1 |
| TAR_db4 | **VolcDB4** | Volcanic emission data set (Diehl et al. 2012) and updates | Transient 1998-2010 time-varying | 13 | 3 |
| TAR_sub | **VolcDBSUB** | subset of strongest 8 volcanoes; averaged SO$_2$ emissions and averaged injection heights from **VolcDB1/2/3** | Transient 1998-2012 monthly-varying | 15 | 2 |
| TAR_db1_3D | VolcDB1_3D | **netCDF** version of volcanic emission data set **VolcDB1** (Bruehl et al., 2015 and updates) | Transient 1998-2012 monthly-varying | 15 | 3 |


**Table 5: Overview of TAR experiments.**





| **Exp- Name** | **Specific description / Volcanic emission** | **Period** | **Ensemble Size** | **Years per member** | **TiER** |
|---|---|---|---|---|---|
| HErSEA_Pin_Em_Ism | Pinatubo episode, SO₂ Emission = medium, Inject shallow @medium-alt. | Transient **1991-1995** incl. GHGs & ODSs (monthly-varying SST & sea-ice from HadISST as for CCMI) | 3 | 5 | 1 |
| HErSEA_Pin_Eh_Ism | Pinatubo episode, SO₂ Emission = high, Inject shallow @medium-alt. | | 3 | 5 | 1 |
| HErSEA_Pin_El_Ism | Pinatubo episode, SO₂ Emission = low, Inject shallow @medium-alt | | 3 | 5 | 1 |
| HErSEA_Pin_Em_Isl | Pinatubo episode, SO₂ Emission = medium, Inject shallow @low-alt | | 3 | 5 | 2 |
| HErSEA_Pin_Em_Idp | Pinatubo episode, SO₂ Emission= medium, Inject over deep altitude-range | | 3 | 5 | 2 |
| HErSEA_Pin_Cntrol | Pinatubo episode, No Pinatubo SO₂ emission | | 3 | 5 | 1 |
| HErSEA_ElC_Em_Ism | El Chichón episode, SO₂ Emission= medium, Inject shallow@ medium-alt | Transient 1982-1986 incl. GHGs & ODSs (monthly-varying SST and sea-ice from HadISST as for CCMI) | 3 | 5 | 1 |
| HErSEA_ElC_Eh_Ism | El Chichón episode, SO₂ Emission= high, Inject shallow@medium-alt | | 3 | 5 | 1 |
| HErSEA_ElC_El_Ism | El Chichón episode, SO₂ Emission = low, Inject shallow@medium-alt | | 3 | 5 | 1 |
| HErSEA_ElC_Em_Isl | El Chichón episode, SO₂ Emission=medium, Inject shallow@low-altitude | | 3 | 5 | 2 |
| HErSEA_ElC_Em_Idp | El Chichón episode, SO₂ Emission= medium, Inject over deep altitude-range | | 3 | 5 | 2 |
| HErSEA_ElC_Cntrol | El Chichón episode no El Chichón SO₂ emission | | 3 | 5 | 1 |
| HErSEA_Agg_Em_Ism | Agung episode SO₂ Emission= medium, Inject shallow @medium-alt | Transient 1963-1967 incl. GHGs & ODSs( monthly-varying SST and sea-ice from HadISST as for CCMI) | 3 | 5 | 1 |
| HErSEA_Agg_Eh_Ism | Agung episode, SO₂ Emission= high, Inject shallow @medium-alt | | 3 | 5 | 1 |
| HErSEA_Agg_El_Ism | Agung episode, SO₂ Emission = low, Inject shallow @medium-alt | | 3 | 5 | 1 |
| HErSEA_Agg_Em_Isl | Agung episode, SO₂ Emission = medium, Inject shallow @low-alt | | 3 | 5 | 2 |
| HErSEA_Agg_Em_Idp | Agung episode, SO₂ Emission =medium, Inject over deep altitude-range | | 3 | 5 | 2 |
| HErSEA_Agg_Cntrol | Agung episode no Agung SO₂ emission | | 3 | 5 | 1 |

**Table 6: Overview of HErSEA experiments**






| Eruption | Measurement/platform | References |
|---|---|---|
| Pinatubo | Extinction/AOD [multi-l]: SAGE-II, AVHRR, HALOE,CLAES | Hamill and Brogniez (SPARC, 2006, and references therein) |
| | Balloon-borne size-resolved concentration profiles (CPC, OPC) | Deshler et al (1994, Kiruna, EASOE), Deshler et al. (2003) |
| | Impactors on ER2 (AASE2), FCAS and FSSP on ER2 (AASE2) | Pueschel et al. (1994), Wilson et al. (1993), Brock et al. (1993) |
| | Ground-based lidar; airborne lidar | NDACC archive; Young, S. A et al. (1994), Browell et al., (1993) |
| | Ship-borne lidar measurements | Avdyushin et al. (1993); Nardi et al. (1993), Stevens et al. (1994) |
| El-Chichón | Satellite extinction/AOD 1000nm (SAM-II) | Hamill and Brogniez (SPARC, 2006 & references therein) |
| | Balloon-borne particle concentration profiles | Hofmann and Rosen (1983; 1987). |
| | Ground-based lidar | NDACC archive |
| Agung | Surface radiation measurements (global dataset gathered in Dyer and Hicks; 1968) | Dyer and Hicks (1965), Pueschel et al. (1972), Moreno and Stock (1964), Flowers and Viebrock (1965) |
| | Balloon-borne measurements | Rosen (1964; 1966, 1968), Pittock (1966) |
| | Ground-based lidar, searchlight and twilight measurements | Clemesha et al. (1966), Grams & Fiocco (1967), Kent et al. (1967)<br>Elterman et al., (1969), Volz (1964; 1965; 1970) |
| | Aircraft measurements | Mossop et al. (1963; 1964), Friend (1966) |

**Table 7 List of stratospheric aerosol observation datasets from the 3 large eruptions of the 21st century (Agung, El**
**Chichón and Mt. Pinatubo). For NDACC archive, see http://www.ndsc.ncep.noaa.gov/data/**



| Eruption | Location | Date | SO₂ (Tg) | Shallow x 2 | Deep |
|---|---|---|---|---|---|
| Mt. Pinatubo | 15°N,120°E | 15/06/1991 | 10-20 (14) | 18-20,21-23km | 18-25km |
| El Chichón | 17°N,93°W | 04/04/1982 | 5-10 (7) | 22-24,24-26km | 22-27km |
| Mt. Agung | 8°S,115°E | 17/03/1963 | 5-10 (7.) | 17-19,20-22km | 17-23km |

**Table 8: Settings to use for initialising the mini-ensemble of interactive stratospheric aerosol simulations for each**
**eruption in the HErSEA experiment. For Pinatubo the upper range of SO₂ emission is based on TOMS/TOVS SO₂**
**observations (Guo et al., 2004a). The SO₂ emissions flux ranges and central-values (in parentheses) are specifically for**
**application in interactive stratospheric aerosol (ISA) models, rather than any new data compilation. the lower range**
**and the central values according to some recent Pinatubo studies (Dhomse et al., 2014; Mills et al., 2016; Sheng et al.,**
**2015a) which have identified a modest downward-adjustment of initial observed SO₂ amounts to agree to**
**HIRS/ISAMS measurements of peak sulphate aerosol loading (Baran and Foot, 1994). The adjustment assumes**
**either uncertainties in the satellite measurements or that loss pathways in the first few weeks after these eruptions are**
**either underpredicted (e.g. due to coarse spatial resolution) or omitted completely (accommodation onto ash/ice) in**
**the ISA models. The El Chichón SO₂ central estimate is taken from Krueger et al. (2008), and an emission range**
**based on assumed ±33% while for Agung the SO₂ emission estimate is from Self and King (1996). For Pinatubo,**
**injection height-ranges for the two shallow and one deep realisation are taken from Antuña et al. (2002). The El**
**Chichón values are based on the tropical lidar signal from Figure 4.34 of Hamill and Brogniez (2006), whereas for**
**Agung we considered the measurements presented in Dyer and Hicks (1968) including balloon soundings (Rosen,**
**1964) and ground-based lidar (Grams and Fiocco, 1967).**






| SO$_2$ mass (Tg S) | Study | SO$_2$ Height (km) |
|---|---|---|
| 5 | Dhomse et al., 2014 | 19-27 |
| 5 | Mills et al. (2016) | 18-20 |
| 7 | Sheng et al. (2015a;b) | 17-30 |
| 8.5 | Timmreck et al. (1999a;b) | 20-27 |
| 8.5 | Niemeier et al. (2009); Toohey et al. (2011) | 24 |
| 8.5 | Brühl et al., (2015) | 18-26* |
| 10 | Pitari and Mancini (2002) | 18-25 |
| 10 | Oman et al. (2006) | 19-29 |
| 10 | Aquila et al. (2012; 2013) | 16-18, 17-27 |
| 10 | English et al. (2013) | 15.1-28.5 |


**Table 9: List of SO$_2$ injection settings used in different interactive stratospheric aerosol model simulations of the 1991**
**Mount Pinatubo eruption. * main peak at 23.5km, secondary peak at 21km.**




| | Parameters | Minimum set | Reduced set | Standard set | Uncertainty range |
|---|---|---|---|---|---|
| 1 | Injected $SO_2$ mass | X | X | X | 5 Tg-S – 10 Tg-S |
| 2 | Mid-point height of 3km-thick injection | X | X | X | 18km – 30km |
| 3 | Latitudinal extent of the injection | X | X | X | Factor 0-1 to vary from 1-box injection at 15N (factor=0) to equator-to-15N (factor=1) * |
| 4 | Sedimentation velocity | | X | X | Multiply model calculated velocity by a factor 0.5 to 2. |
| 5 | $SO_2$ oxidation scaling | | X | X | Scale gas phase oxidation of $SO_2$ by a factor 0.5 to 2 |
| 6 | Nucleation rate of sulfate particles | | | X | Scale model calculated rate by a factor 0.5 to 2. |
| 7 | Sub-grid particle formation factor. | | | X | Emit fraction of $SO_2$ as sulphuric acid particles formed at sub-grid-scale (0 to 10%) |
| 8 | Coagulation rate | | | X | Scale the model calculated rate by a factor 0.5 to 2. |

**Table 10: Groups will need to translate the 0-1 latitude-spread parameter into a sequence of fractional injections into**
**all grid boxes between the equator and 15 °N. For example for a model with 2.5 degree latitude resolution, the**
**relative injection in the 6 latitude bins between 0 and 15N would take the form [0,0,0,0,0,1] for extent factor=0, and**
**[0.167,0.167, 0.167,0.167, 0.167,0.167] for extent factor=1. Injection ratios for intermediate values of the spread factor**
**would be calculated by interpolation between these two end member cases.**





| Exp- Name | Specific description / Volcanic emission | Period | TIER |
|---|---|---|---|
| PoEMS_OAT_med | SO$_2$ Emission = medium, Inject shallow @medium-alt. Processes unperturbed. | Transient 1991-1995 | 1 |
| PoEMS_OAT_P4h | SO$_2$ Emission = medium, Inject shallow @medium-alt. Sedimentation rates doubled | | 2 |
| PoEMS_OAT_P4l | SO$_2$ Emission = medium, Inject shallow @medium-alt. Sedimentation rates halved | | 2 |
| PoEMS_OAT_P5h | SO$_2$ Emission = medium, Inject shallow @medium-alt. SO$_2$ oxidation rates doubled | | 3 |
| PoEMS_OAT_P5l | SO$_2$ Emission = medium, Inject shallow @medium-alt. SO$_2$ oxidation rates halved | | 3 |
| PoEMS_OAT_P6h | SO$_2$ Emission = medium, Inject shallow @medium-alt. Nucleation rates doubled | | 3 |
| PoEMS_OAT_P6l | SO$_2$ Emission = medium, Inject shallow @medium-alt. Nucleation rates halved | | 3 |
| PoEMS_OAT_P7h | SO$_2$ Emission = medium, Inject shallow @medium-alt. % SO$_2$ as primary SO$_4$ x2 | | 3 |
| PoEMS_OAT_P7l | SO2 Emission = medium, Inject shallow @medium-alt. % SO$_2$ as primary SO$_4$ x0.5 | | 3 |
| PoEMS_OAT_P8h | SO$_2$ Emission = medium, Inject shallow @medium-alt. Coagulation rates doubled | | 2 |
| PoEMS_OAT_P8l | SO$_2$ Emission = medium, Inject shallow @medium-alt. Coagulation rates halved | | 2 |


**Table 11: Overview of PoEMS One-At-a-Time" (OAT) test runs. Note that when imposing the parameter-scaling, the**
**models should only enact the change in volcanically-enhanced air masses (where the total sulphur volume mixing**
**ratio exceeds a threshold suitable for their model). Perturbing only the volcanically-enhanced air masses will ensure,**
**pre-eruption conditions and tropospheric aerosol properties remains unchanged by the scalings.**




Figures

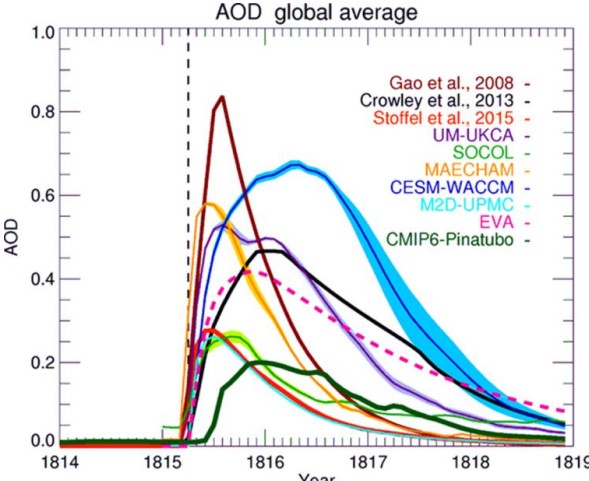

**Figure 1 Uncertainty in estimates of radiative forcing parameters for the 1815 eruption of Mt. Tambora: Global-**
**average aerosol optical depth (AOD) in the visible band from an ensemble of simulations with chemistry–climate**
**models forced with a 60 Tg SO₂ equatorial eruption, from the Easy Volcanic Aerosol (EVA, Toohey et al., 2016b)**
**module with 56.2 Tg SO₂ equatorial eruptions (magenta thick dashed line), from Stoffel et al. (2015), from Crowley**
**and Unterman (2013), and from Gao et al. (2008, aligned so that the eruption starts on April 1815). The estimate for**
**the Pinatubo eruption as used in the CMIP6 historical experiment is also reported for comparison. The black triangle**
**shows latitudinal position and timing of the eruption. Chemistry–climate models are CESM (WACCM) (Mills et al.,**
**2016), MAECHAM5-HAM (Niemeier et al., 2009), SOCOL (Sheng et al., 2015a), UM-UKCA (Dhomse et al., 2014),**
**and CAMB-UPMC-M2D (Bekki, 1995; Bekki et al., 1996). For models producing an ensemble of simulations, the line**
**and shading are the ensemble mean and ensemble standard deviation respectively. Figure from Zanchettin et al.**
**(2016).**



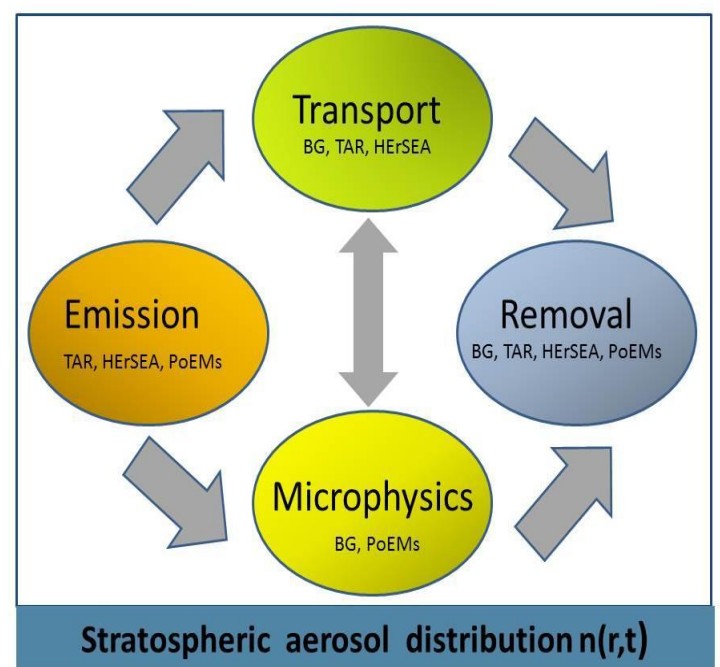



**Figure 2 Schematic overview over the processes that influence the stratospheric aerosol size distribution. The related SSiRC experiments are listed below. BG stands for "BackGround", TAR for "Transient Aerosol Record", HErSEA for "Historical Eruption SO$_2$ Emission Assessment"andPoEMs for "Pinatubo Emulation in Multiple models".**






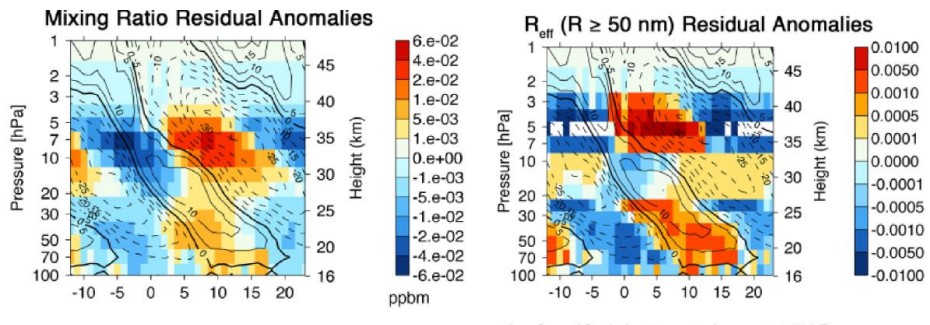


**Figure 3.** **(a) Composite of QBO-induced residual anomalies in the MAECHAM5-SAM2 modelled aerosol mass mixing ratio with respect to the time of onset of westerly zonal mean zonal wind at 18 hPa. Black contours denote the residual zonal wind. Dashed lines represent easterlies, contour interval is 5ms (b) same but for the modelled effective radius of aerosols with R≥50 nm. Figure from Hommel et al. (2015).**




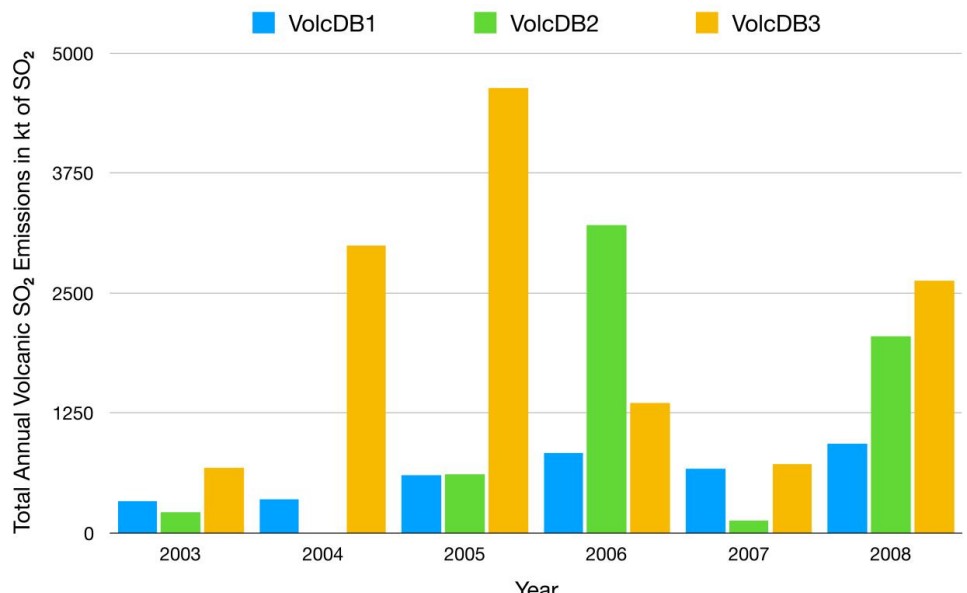

**Figure 4: Annual total volcanic sulfur dioxide (SO$_2$) emission from three different emission data sets between 2003**
**and 2008 to be used in the TIER1 MITAR experiments. VolcDB1 (Bingen et al., 2017) considers only stratospheric**
**SO$_2$ emissions, VolcDB2( Neely and Schmidt, 2016) and VolcDB3 (Carn et al., 2016) consider both tropospheric and**
**stratospheric SO$_2$ emission.**




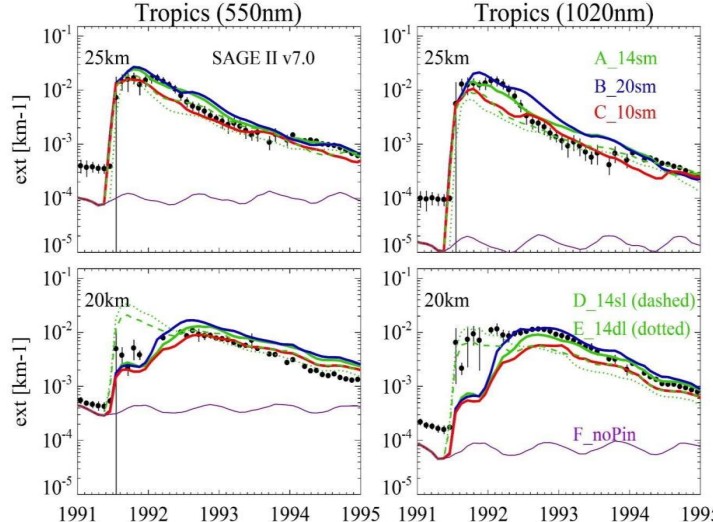


**Figure 5: Example results from interactive stratospheric aerosol simulations with the UM-UKCA model (Dhomse et al., 2014) of 5 different SO₂-injection-realisations of the 1991 Pinatubo eruption (see Table 3.3.1), The model tropical –mean extinction in the mid-visible (550nm) and near-infra-red (1020nm) is compared to that from SAGE-II measurements. Only 2 of the 5 injection realisations inject below 20km and the impact on the timing of the peak, and general evolution of the aerosol optical properties is apparent. In this model the growth to larger particle sizes and subsequent sedimentation to lower altitudes is able to explain certain signatures seen in the satellite data (see also Mann et al., 2015).**





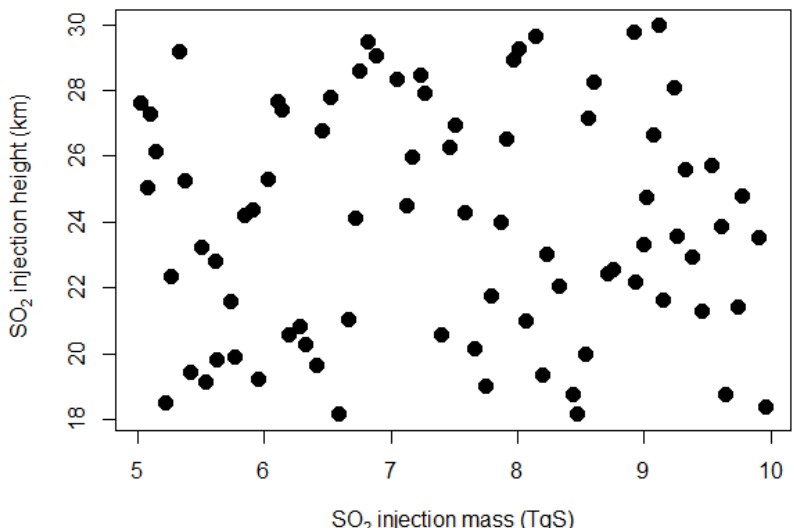


**Figure 6 Illustration of the latin hypercube sampling method. Each dot represents the value used in one of the particular simulations with a perturbed parameter ensemble (PPE) with 50 members (realisations/integrations).**



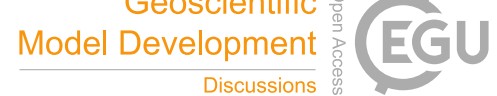



**List of Abbreviations**

| | |
|---|---|
| AEROCOM | Aerosol Comparisons between Observations and Models |
| AOD | Aerosol Optical Depth |
| AMOC | Atlantic Meridional Overturning Circulation |
| ASAP2006 | Assessment of Stratospheric Aerosol properties (WMO, 2006) |
| AVHRR | Advanced Very High Resolution Radiometer |
| BDC | Brewer-Dobson Circulation |
| CALIOP | Cloud-Aerosol Lidar with Orthogonal Polarization |
| CALIPSO | Cloud-Aerosol Lidar and Infrared Pathfinder Satellite Observations |
| CATS | Cloud-Aerosol Transport System |
| CCM | Chemistry Climate Model |
| CCMVal | Chemistry-Climate Model Validation Activity |
| CCMI | Chemistry-Climate Model Initiative |
| CCN | Cloud Condensation Nuclei |
| CDN | Cloud Droplet Number Concentration |
| CDR | Cloud Droplet Radius |
| CMIP | Coupled Model Intercomparison Project |
| CMIP5 | Coupled Model Intercomparison Project, phase 5 |
| CMIP6 | Coupled Model Intercomparison Project, phase 6 |
| DJF | December-January-February |
| DWD | Deutscher Wetterdienst |
| ECHAM | European Center/HAMburg model, atmospheric GCM |
| EGU | European Geophysical Union |
| ECMWF | European Centre for Medium-Range Weather Forecasting |
| EESC | Equivalent Effective Stratospheric Chlorine |
| ENSO | El Niño Southern Oscillation |
| ENVISAT | Environmental Satellite |
| ERA-Interim | ECMWF Interim Re-Analysis |
| ERBE | Earth Radiation Budget Experiment |
| ESA | European Space Agency |
| ESM | Earth System Model |
| EVA | Easy Volcanic Aerosol |
| GCM | General Circulation Model |
| GHG | Green House Gases |
| GOMOS | Global Ozone Monitoring by Occultation of Stars |
| HALOE | Halogen Occultation Experiment |
| HD(CP)2 | High definition clouds and precipitation for advancing climate prediction |
| ISA-MIP | Interactive Stratospheric Aerosol Model Intercomparion Project |
| ICON | ICOsahedral Nonhydrostatic |
| IPCC | Intergovernmental Panel on Climate Change |
| ISCCP | International Satellite Cloud Climatology Project (ISCCP) |
| ITCZ | Intertropical Convergence Zone |
| JAXA | Japanese Aerospace Exploration Agency |



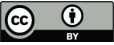

| JJA | June-July-August |
| LAI | Leaf Area Index |
| LW | Longwave |
| LWP | Liquid Water Path |
| MiKlIP | Mittelfristige Klimaprognosen |
| MIPAS | Michelson Interferometer for Passive Atmospheric Sounding |
| MODIS | Moderate Imaging Spectroradiometer |
| MPI-ESM | Earth System model of Max Planck Institute for Meteorology |
| NAO | North Atlantic Oscillation |
| NH | Northern hemisphere |
| OLR | Outgoing longwave radiation |
| OMI | Ozone Monitoring Instrument |
| OMPS | Ozone Mapping and Profiler Suite |
| OMPS-LP | Ozone Mapping and Profiler Suite–Limb Profiler |
| OPC | Optical Particle Counter |
| OSIRIS | Optical Spectrograph and InfraRed Imager System |
| PDF | Probability Density Function |
| POAM | Polar Ozone and Aerosol MeasurementPSD |
| PSD | Particle Size Distribution |
| QBO | Quasi-biennial oscillation |
| RF | Radiative Forcing |
| RH | Relative Humidity |
| SAOD | Stratospheric Aerosol Optical Depth |
| SAGE | Stratospheric Aerosol and Gas Experiment |
| SAM | Southern Annular Mode |
| SCIAMACHY | Scanning Imaging Absorption Spectrometer for Atmospheric Chartography |
| SH | Southern Hemisphere |
| SPARC | Stratosphere-troposphere Processes And their Role in Climate |
| SSiRC | Stratospheric Sulfur and its Role in Climate |
| SST | Sea Surface Temperature |
| SW | Shortwave |
| TCS | Transient Climate Sensitivity |
| ToA | Top of the Atmosphere |
| TOMS | Total Ozone Mapping Spectrometer |
| TOVS | TIROS Operational Vertical Sounder |
| VEI | Volcanic Explosivity Index |
| VolMIP | Model Intercomparison Project on the climate response to Volcanic forcing |

