# Peer review of "The Interactive Stratospheric Aerosol Model Intercomparison"

_Geoscientific Model Development, 2017_

## Editor Comment (EC1) · S. Bekki (Editor) · 3 Feb 2018

There are various tables providing boundary conditions for emissions. The authors should provide them under the form of ascii files with a link. That way, modellers who want to run the experiments just have to download the input files instead of having to recreate the files from the tables.

---

## Referee Comment (RC1) · Anonymous Referee #1 · 5 Feb 2018

Review of the paper: *The Interactive Stratospheric Aerosol Model Intercomparison Project (ISA-MIP): Motivation and experimental design*, by C. Timmreck et al., Geosci. Model Dev. Discuss., https://doi.org/10.5194/gmd-2017-308, 2018.

This paper presents a new Model Intercomparison Project, that will result very useful for the stratospheric aerosol-chemistry-climate community. Motivation and protocols for input/output data relative to four different experiments are presented in Section 3 in a detailed and clear way, so that (once published) this paper can be used in the community as a guideline for the experiments to be completed by the modellers.

The introductory part to the actual MIP description (i.e., Sections 1-2) is also well written and clear and help the reader capturing the importance for a better understanding of stratospheric aerosols, their variability, role of gas precursors, climate impact of stratospheric sulfate from explosive volcanic eruptions, and other important aspects (large scale transport, QBO, size distribution, etc).

Here, however, the manuscript may be somewhat improved (in my opinion), in at least two main aspects, discussed below. For this reason, I recommend publication of this paper, after these specific points are properly addressed.

*Specific points*

(1) The authors discuss the climate impact of stratospheric volcanic aerosols, how their large scale distribution may be affected by stratospheric transport oscillations (QBO) and how their size distribution may change as a function of the injected $SO_2$. A paragraph should be added, addressing the potential impact of the aerosol radiative interactions on some features of stratospheric dynamics and transport, as age of air and strat-trop exchange of trace species. Recent studies which may be relevant from this point of view, are those by Ray et al. (2014), Pitari et al. (2016a), Diallo et al. (2017). A brief paragraph on this aspect would make even stronger the need for the proposed MIP. This paragraph could probably be inserted in the Introduction or at the end of Subsection 3.3.2.

Ray, E.A. et al.: Improving stratospheric transport trend analysis based on $SF_6$ and $CO_2$ measurements, *J. Geophys. Res*., *119*, doi:10.1002/2014JD021802, 2014.

Pitari, G., et al.: Impact of stratospheric volcanic aerosols on age-of-air and transport of long-lived species, *Atmosphere, 7,* 149, doi: 10.3390/atmos7110149, 2016a.

Diallo, M., et al.: Significant Contributions of Volcanic Aerosols to Decadal Changes in the Stratospheric Circulation, *Geophys. Res. Lett., 44*, 10780, https://doi.org/10.1002/2017GL074662, 2017.

(2) References to new studies on volcanic aerosols may be added. The QBO impact on aerosol dispersal and e-folding time has been discussed in Pitari et al. (2016b) and could be cited at page 5 line 181. A re-examination of the initial $SO_2$ cloud lifetime was made in Mills et al. (2017) and could be cited at page 2 line 51.

Pitari, G., et al.: Stratospheric aerosols from major volcanic eruptions: a composition-climate model study of the aerosol cloud dispersal and *e*-folding time, *Atmosphere*, *7,* 75, doi:10.3390/atmos7060075, 2016b.

Mills, M.J., et al.: Radiative and chemical response to interactive stratospheric sulfate aerosols in fully coupled CESM1(WACCM), *J. Geophys. Res.,* doi: 10.1002/2017JD027006, 2017.

(3) At page 7 lines 271-273 the authors write: "Modelling groups are encouraged to include a set of passive tracers to diagnose the atmospheric transport independently from emissions mostly following the CCMI recommendations (Eyring et al., 2013). These tracers are listed in Table S3 in the supplementary material." It should be specified that in case modelling groups had already run these experiments, results produced and uploaded for CCMI may also be used for ISA-MIP, taking them directly from the CCMI data repository. I would also suggest to provide a link (as made in Eyring et al., 2013) where gridded input data may be available for download (S fluxes etc.).

*Minor points*

At the beginning of Section 2 (page 5 lines 15-155) the following sentence sounds odd: "However, the focus of the ISA-MIP experiments described here is on comparing to measurements of the overall optical and physical properties of the stratospheric aerosol layer, which is manly determined by stratospheric aerosol". Maybe the final "aerosol" should be substituted with sulfate.

The discussion at the end of Section 2 (page 7 lines 210-219) could probably be made even more robust with reference to sulfate geoengineering studies. Some of these have highlighted differences in what the authors themselves call "a crucial point", i.e., the different degree of isolation of the tropical pipe and the meridional transport of sulfate aerosols through the subtropical barrier. See for example Tilmes et al. (2015) and Visioni et al. (2018).

Tilmes, S., et al.: A new Geoengineering Model Intercomparison Project (GeoMIP) experiment designed for climate and chemistry models, *Geosci. Model Dev., 8,* 43-49, doi: 10.5194/gmd-8-43-2015, 2015.

Visioni, D., et al.: Sulfur deposition changes under sulfate geoengineering conditions: QBO effects on transport and lifetime of stratospheric aerosols, *Atmos. Chem. Phys., in press,* 2018. [*Atmos. Chem. Phys. Discuss.*, https://doi.org/10.5194/acp-2017-987, 2017; accepted for publication on Atmos. Chem. Phys.].

---

## Referee Comment (RC2) · Anonymous Referee #2 · 11 Feb 2018

This paper motivates and describes the experimental design of ISA-MIP, the Interactive Stratospheric Aerosol Model Intercomparison Project, which is part of the SPARC-SSiRC initiative.

I found ISA-MIP very well motivated and structured. It builds on four experiments that are designed to tackle different processes involved in the formation and development of stratospheric aerosol. ISA-MIP has a strong orientation toward comparison of the model results with instrumental observations and promises to deliver key information to improve our understanding of the stratospheric aerosol layer.

The paper certainly fits the scope of GMD. It is well written and the technical description

of the four experiments is overall clear and precise in its different aspects. I have a few suggestions and requests for clarification that do not detract from the fact that I recommend this paper for publication.

My only comment regarding the protocol concerns the TAR experiment "db1". This is a mandatory (Tier 1) experiment that uses preferably tabulated point sources or, optionally, 3D data. However, the latter option is given an own identification in the protocol and a different priority (Tier 3). There may thus be a conflict, as a mandatory experiment is optionally bypassed by performing a low-priority experiment. If the two experiments are equivalent alternatives, they should be given the same priority, or even appear as the same experiment with the selected option to be reported in the metadata. I recommend some clarification.

As a general note, an expanded description about potential synergies and links with other ongoing MIPs would better highlight the value of ISA-MIP for the broader climate modelling community.

Specific comments on the manuscript:

Line 78: please check, the acronym OCS seems to be only introduced in line 164

Line 186: "compared to moderate eruptions"

Line 295: Across?

Line 321: the nudging period for the QBO is 1980-2000 (21 years) but the experiment only consists of 20 years. It seems that to include the year 2000 at the end of the simulation, the nudging period should start in 1981.

Paragraph 3.3.3: It appears from Tables 5 and 6 that "VolcDB*" identify the datasets, whereas the experiment names are "TAR_db*/TAR_sub". It seems that the text in this paragraph mixes the two (for instance in lines 374-376)

Lines 432-438: this is certainly an interesting goal, but in this short description this

[Figure]

appears at the edge or even slightly out of the scope of ISA-MIP itself. Can you expand on this?

Table 1: some of the information provided is not clearly described. For instance, are the numbers in parentheses in the "Total years" column the recommended integration years? This seems not to hold for the PoEMS where the numbers seem to refer to the number of perturbed parameters. The description of the number of specific experiments for PoEMS also seems to lack clarity.

---

## Author Comment (AC1) · 17 Apr 2018

**Response to reviewer 2**

Thank you very much, for your very helpful comments and suggestions (indicated in bold and italic). You will find our point-by-point reply to them below.

*"My only comment regarding the protocol concerns the TAR experiment "db1". This is a mandatory (Tier 1) experiment that uses preferably tabulated point sources or, optionally, 3D data. However, the latter option is given an own identification in the protocol and a different priority (Tier 3). There may thus be a conflict, as a mandatory experiment is optionally bypassed by performing a low-priority experiment. If the two experiments are equivalent alternatives, they should be given the same priority, or even appear as the same experiment with the selected option to be reported in the metadata.*

*I recommend some clarification."*

We accept the original description of the TAR experiment may have been confusing. To ensure comparability between all of the three data sets we agreed on point sources for the volcanic emission in the mandatory (TIER1) experiments. For clarification we have changed the following sentence:

Page 11, lines 379-380: "If modelling groups prefer not to use point sources, we additionally offer VolcDB1_3D which provides a series of discrete 3D gridded $SO_2$ injections at specified times."

to

Page 11, lines 37-382: "To test the effect of the implementation strategy (point source vs cloud) an additional non-mandatory experiment has been set up: tar_db1_sub with VolcDB1_3D as corresponding data set which provides a series of discrete 3D gridded $SO_2$ injections at specified times. "

To clarify, we also changed slightly the text:

Page 12, lines 401-402: "The optionally provided VolcDB1_3D data set, contains volume mixing ratio distributions of the injected $SO_2$ on a T42 Gaussian grid with 90 levels.

to

Page 12, lines 401-404: "The VolcDB1_3D data set, for the optional experiment tar_db1_3D contains volume mixing ratio distributions of the injected $SO_2$ cloud on a T42 Gaussian grid with 90 levels. The integral $SO_2$ mass for each injection is the same."

*"As a general note, an expanded description about potential synergies and links with other ongoing MIPs would better highlight the value of ISA-MIP for the broader climate modelling community."*

We have included in the summary the following sentences:

Pages 18-19, lines 632-649: "For example, the CMIP6 Geoengineering Model Intercomparison Project (GeoMIP, Kravitz et al., 2015) investigates common ways in which climate models treat various geoengineering scenarios some of them via sulphate aerosols (e.g. Tilmes et al., 2015). However, there is a large inter model spread for the cooling efficiency of sulphate aerosol, i.e. the normalized cooling rate per injected unit of sulphur (Moriyama et al., 2016). ISA-MIP is therefore of special importance for GeoMIP as it could help to understand the reason for these uncertainties, to better constrain the forcing efficiency and to improve future scenarios. Furthermore it is so far not clear whether the large inter-model spread of the CMIP5 models in the simulated

post-volcanic climate response mostly depends on uncertainties in the imposed volcanic forcing or on an insufficient representation of climate processes. To discriminate the individual uncertainty factors it is useful to develop standardized experiments/model activities that systematically address specific uncertainty factors. Hence ISA-MIP, which covers the uncertainties in the pathway from the eruption source to the volcanic radiative forcing, will complement the CMIP6 VolMIP project (Zanchettin et al., 2016) which addresses the pathway from the forcing to the climate response and the feedback, by studying the uncertainties in the post-volcanic climate response to a well-defined volcanic forcing. ISA-MIP also complements the chemistry climate model initiative CCMI (Eyring et al., 2013) and the Aerosol Comparison (AeroCom) initiative (Schulz et al., 2006) as well as the Aerosol Chemistry Model Intercomparison Project (AerChemMIP, Collins et al., 2017) as it concentrates on stratospheric aerosol which is not in the focus of all these activities."

***Specific comments on the manuscript:***

***Line 78: please check, the acronym OCS seems to be only introduced in line 164***

We have included the explanation of OCS now earlier in the manuscript.

Page 3, lines 90-91: "we now have a 2002-2012 long record of global altitude-resolved $SO_2$, and carbonyl sulphide (OCS) and aerosol …"

*Line 186: "compared to moderate eruptions"*

We have revised the sentence accordingly to:

Page 6, lines 200-201: "… and predict a reduced cooling efficiency compared to moderate eruptions with moderate sulphur injections (e.g. Timmreck et al., 2010; English et al., 2013).

***"Line 295: Across?"***

We have revised the sentence to:

Page 9, line 313: "although new in situ measurements indicate that the cross-tropopause-SO2-flux is neglible over Mexico and central America (Rollins et al., 2017)."

***"Line 321: the nudging period for the QBO is 1980-2000 (21 years) but the experiment only consists of 20 years. It seems that to include the year 2000 at the end of the simulation, the nudging period should start in 1981.***"

We have revised this accordingly:

Page 10, lines 329-340: .Modelling groups should run this simulation with varying QBO, either internally generated or nudged to the 1980-2000 period.

***"Paragraph 3.3.3: It appears from Tables 5 and 6 that "VolcDB*" identify the datasets, whereas the experiment names are "TAR_db*/TAR_sub". It seems that the text in this paragraph mixes the two (for instance in lines 374-376)."***

Thank you very much for this hint. We have revised the sentences to

Page 12, lines 394-397: "Summarising the number of experiments to be conducted within TAR: four are mandatory (TAR_base with no volcanic emission, Tar_db1/2/3), one additional is recommended (TAR-sub) and two others are optional (TAR_db4 and TAR_db1_3D; see Table 5 for an overview)."

*Lines 432-438: this is certainly an interesting goal, but in this short description this appears at the edge or even slightly out of the scope of ISA-MIP itself. Can you expand on this?*

Whilst we agree with the reviewer that there is no specific experiment aiming to understand the relationship between the ice core sulphate deposition and the stratospheric aerosol layer enhancements that drives the radiative cooling, the idea was to suggest that there is the potential for a systematic multi-model analysis of those 2 metrics (based on the HErSEA results) and seek to identify how uncertain historic volcanic forcings derived from ice core sulphate deposition may be.

We have added the following sentence to the revised manuscript stating that:

Page 13-14, lines, 440-463:"Although HErSEA has no specific experiment to understand the relationship between the ice core sulphate deposition and the stratospheric aerosol layer enhancements that drive the surface cooling, there is the potential for a systematic inter-model study (e.g. similar to Marshall et al., 2018) to identify how uncertain historic volcanic forcings derived from ice core sulphate deposition may be."

*"Table 1: some of the information provided is not clearly described. For instance, are the numbers in parentheses in the "Total years" column the recommended integration years? This seems not to hold for the PoEMS where the numbers seem to refer to the number of perturbed parameters. The description of the number of specific experiments for PoEMS also seems to lack clarity."*

There was a mistake in the Table provided in the Discussions version of this article which we agree was confusing. In the revised manuscript we have changed Table 1 to be as shown below. We have also re-iterated (in the section 3.4.2 of the text, and in Table 1) the important requirement (currently only explained in the caption to Table 11) that the PoEMS parameter-scalings must only be applied in gridboxes with "volcanically-enhanced airmasses" (determined either by total-sulphur-vmr-threshold or the "passive Volc tracer".

Page 17, lines 592-597: "When imposing the parameter-scalings, the models must only enact that change in grid boxes with volcanically-enhanced air masses. This can be determined either via total sulphur volume mixing ratio threshold suitable for the particular model, or via the "passive tracer Volc" recommended in section 3.3.3. Restricting the perturbation to the Pinatubo sulphur will leave pre-eruption conditions and tropospheric aerosol properties unchanged. ensuring a clean "uncertainty pdf" for the volcanic forcing."

[revised manuscript text omitted]

---

## Author Comment (AC2) · 17 Apr 2018

**Response to reviewer 1**

Thank you very much, for your very helpful comments and suggestions (indicated in bold and italic). You will find our point-by-point reply to them below.

***"The authors discuss the climate impact of stratospheric volcanic aerosols, how their large scale distribution may be affected by stratospheric transport oscillations (QBO) and how their size distribution may change as a function of the injected SO2. A paragraph should be added, addressing the potential impact of the aerosol radiative interactions on some features of stratospheric dynamics and transport, as age of air and strat-trop exchange of trace species. Recent studies which may be relevant from this point of view, are those by Ray et al. (2014), Pitari et al. (2016a), Diallo et al. (2017). A brief paragraph on this aspect would make even stronger the need for the proposed MIP. This paragraph could probably be inserted in the Introduction or at the end of Subsection 3.3.2."***

As the reviewer suggested we have included a couple of sentences on stratospheric aerosol and dynamics in the introduction and we also discuss uncertainties in mean age of air at the end of section 3.3.2.

Page 2, lines 57-65: "The consequent heating of the stratospheric aerosol layer strongly influences stratospheric dynamics amplifying the Brewer-Dobson circulation (BDC) and modifying the equator-to-pole temperature gradient. These two primary drivers cause changes to geostrophic zonal winds and the propagation of atmospheric waves (e.g. Bittner et al., 2016; Toohey et al., 2014) and lead to a strengthening of the polar vortex (e.g. Charlton-Perez et al., 2013). The heating from continued $SO_2$ injection to the stratosphere may further disturb or even "shut down" the quasi biennial oscillation (QBO) (e.g. Aquila et al., 2014; Niemeier and Schmidt, 2017). These composition-dynamics interactions also influence the transport and residence time of other long-lived species ($N_2O$, $CH_4$) (Pitari et al., 2016a; Visioni et al., 2017). The enhanced stratospheric aerosol layer after large volcanic eruptions causes also large mean age of air variations on time scales of several years (e.g. Ray et al., 2014; Muthers et al., 2016, Garfinkel et al., 2017)."

Page3, lines 78-81: "…..counteracted the warming due to increased greenhouse gases over that period (e.g. Solomon et al., 2011; Ridley et al., 2014; Santer et al., 2015). Small to moderate volcanic eruptions after 2008 also show an impact on the stratospheric circulation in the Northern Hemisphere, in particular on the pattern of decadal mean age variability and its trends during 2002–2011 (Diallo et al., 2017)."

Page 14, lines 487-497: "Analysing how the vertical profile of the enhanced stratospheric aerosol layer evolves during global dispersion and decay, will provide a key indicator for why the models differ, and what are the key driving mechanisms. Furthermore, the actual response of the BDC and mean age of air to Pinatubo is poorly constrained by existing reanalysis data (Garfinkel et al., 2017). While some modeling studies reported a decreasing mean age of air following volcanic eruptions throughout the stratosphere (Garcia et al., 2011; Garfinkel et al., 2017), show other studies an increase in mean age (Diallo et al., 2017). Moreover, Muthers et al. (2016) found decreasing age of air in the middle and upper stratosphere and increasing mean age below, while Pitari et al. (2016a) found decreasing mean age at higher levels of 30 hPa in the tropics and 10 hPa in the middle latitudes after the Pinatubo eruption. The HerSEA experiment in combination with a passive volcanic tracer might therefore help to better constrain the response of the BDC to volcanic eruptions using observations and help to clarify the uncertainties in age of air changes after the Pinatubo eruption. For all three major eruptions, we have identified key observational datasets (Table 7) that will provide benchmark tests to evaluate the vertical profile, covering a range of different aerosol metrics."

***"References to new studies on volcanic aerosols may be added. The QBO impact on aerosol dispersal and e-folding time has been discussed in Pitari et al. (2016b) and could be cited at***

*page 5 line 181. A re-examination of the initial SO2 cloud lifetime was made in Mills et al. (2017) and could be cited at page 2 line 51".*

To take into account new developments/studies we have included a couple of recent published papers (indicated in blue) in the field:

Page 2, lines 59-52: "Major volcanic eruptions inject vast amounts of $SO_2$ into the stratosphere, which is converted into sulphuric acid aerosol with an e-folding time of about a month, which might be prolonged due to OH depletion within the dense $SO_2$ cloud in the first weeks following a large volcanic eruption (Mills et al., 2017)".

Page 3, lines 90- 93: "For example, we now have a 2002-2012 long record of global altitude-resolved $SO_2$ ,and carbonyl sulphide (OCS) and aerosol volume density measurements provided by the Michelson Interferometer for Passive Atmospheric Sounding Environmental Satellite (MIPAS Envisat, Höpfner et al., 2013; 2015; Glatthor et al., 2015, Günther et al., 2018)."

Page 6, lines 193-196: "Internal variability associated with the QBO alters the isolation of the tropical stratosphere and subsequently the poleward transport of tropical stratospheric aerosol, thereby modulating its global dispersal, particle size distribution, and residence time (e.g. Trepte and Hitchmann, 1992; Hommel et al., 2015, Pitari et al., 2016b,)."

See also further new references included in the answers to other points by the reviewer.

*"At page 7 lines 271-273 the authors write: "Modelling groups are encouraged to include a set of passive tracers to diagnose the atmospheric transport independently from emissions mostly following the CCMI recommendations (Eyring et al., 2013). These tracers are listed in Table S3 in the supplementary material." It should be specified that in case modelling groups had already run these experiments, results produced and uploaded for CCMI may also be used for ISA-MIP, taking them directly from the CCMI data repository. I would also suggest to provide a link (as made in Eyring et al., 2013) where gridded input data may be available for download (S fluxes etc.)."*

Thank you very much for your suggestion. This is a good point. We agree it will be valuable to compare the temporal variations in the ISA-MIP experiments (in particular the transient TAR and HErSEA experiments) with those from the CCMI REF-C1 and REF-C1SD simulations, which include the full fix of external forcing variations. Although we do not feel there is a need to specify this in the paper we will certainly encourage the leads for those two ISA-MIP experiments to consider this, and approach the relevant CCMI coordinators accordingly. In particular, there may still be time to double-check whether any extra diagnostics should be added to the ISA-MIP simulations as most groups will still be finalizing the final set-up for their integrations.

Forcings and other data sets will be made available on the ISA-MIP website: http://www.isamip.eu and through specific links, which will be included in the revised manuscript.

*"At the beginning of Section 2 (page 5 lines 15-155) the following sentence sounds odd: "However, the focus of the ISA-MIP experiments described here is on comparing to measurements of the overall optical and physical properties of the stratospheric aerosol*

*layer, which is manly determined by stratospheric aerosol". Maybe the final "aerosol" should be substituted with sulfate."*

We have corrected the sentence to:

Page 5, lines 167-169: "However, the focus of the ISA-MIP experiments described here is on comparing to measurements of the overall optical and physical properties of the stratospheric aerosol layer, which is mainly determined by stratospheric aerosol sulphate."

*"The discussion at the end of Section 2 (page 7 lines 210-219) could probably be made even more robust with reference to sulfate geoengineering studies. Some of these have highlighted differences in what the authors themselves call "a crucial point", i.e., the different degree of isolation of the tropical pipe and the meridional transport of sulfate aerosols through the subtropical barrier. See for example Tilmes et al. (2015) and Visioni et al. (2018)."*

We have added a sentence to refer to sulphate geoengineering studies

[revised manuscript text omitted]

---

## Author Comment (AC3) · 17 Apr 2018

**Response to the topical editor (Slimane Bekki)**

*There are various tables providing boundary conditions for emissions. The authors should provide them under the form of ascii files with a link. That way, modellers who want to run the experiments just have to download the input files instead of having to recreate the files from the tables.*

Dear Dr. Bekki,

Thank you very much for your comment. We will include links to all input fields in the revised manuscript and on our web page http://www.isamip.eu . This should ensure that all modellers could easily download the data in either ASCII or NETCDF format.